# Why Do Unlearnable Examples Work: A Novel Perspective of Mutual Information

**Yifan Zhu**[1,2*], **Yibo Miao**[1,2*†], **Yinpeng Dong**[3 4], **Xiao-Shan Gao**[1,2†]

[1]State Key Laboratory of Mathematical Sciences, Academy of Mathematics and Systems Science, Chinese Academy of Sciences
[2]University of Chinese Academy of Sciences
[3]College of AI, Tsinghua University
[4]Shanghai Qi Zhi Institute
zhuyifan@amss.ac.cn, miaoyibo@amss.ac.cn,
dongyinpeng@mail.tsinghua.edu.cn, xgao@mmrc.iss.ac.cn

## Abstract

The volume of freely scraped data on the Internet has driven the tremendous success of deep learning. Along with this comes the growing concern about data privacy and security. Numerous methods for generating unlearnable examples have been proposed to prevent data from being illicitly learned by unauthorized deep models by impeding generalization. However, the existing approaches primarily rely on empirical heuristics, making it challenging to enhance unlearnable examples with solid explanations. In this paper, we analyze and improve unlearnable examples from a novel perspective: *mutual information reduction*. We demonstrate that effective unlearnable examples always decrease mutual information between clean features and poisoned features, and when the network gets deeper, the unlearnability goes better together with lower mutual information. Further, we prove from a covariance reduction perspective that minimizing the conditional covariance of intra-class poisoned features reduces the mutual information between distributions. Based on the theoretical results, we propose a novel unlearnable method called **Mutual Information Unlearnable Examples (MI-UE)** that reduces covariance by maximizing the cosine similarity among intra-class features, thus impeding the generalization effectively. Extensive experiments demonstrate that our approach significantly outperforms the previous methods, even under defense mechanisms.

## 1 Introduction

Deep neural networks (DNNs) have achieved unprecedented performance across various fields in the past decade (LeCun et al., 2015). This progress is largely driven by the availability of large-scale datasets freely scraped from the Internet, such as ImageNet (Deng et al., 2009) and LAION-5B (Schuhmann et al., 2022), which continue to advance state-of-the-art deep models (Achiam et al., 2023; Liu et al., 2024d). However, a concerning fact is that some of this data collection occurs without authorization (Birhane and Prabhu, 2021). Users may be reluctant to contribute their privacy-sensitive data, such as face images and medical reports, to training large-scale commercial models (Achiam et al., 2023; Team et al., 2023; Liu et al., 2024d). Indeed, according to a report (Hill, 2020), a tech company illicitly acquired over three billion facial images to develop a commercial facial recognition model. Other investigations revealed an increasing number of lawsuits between data owners and machine learning companies (Vincent, 2019; Burt, 2020; Conklin, 2020; Dunn, 2024). Consequently, there is a growing emphasis on safeguarding data from unauthorized use for training.

Tremendous efforts have been made to craft unlearnable examples (UEs) in order to prevent data from being illicitly learned by unauthorized deep models (Feng et al., 2019; Huang et al., 2020; Fowl et al., 2021; Sandoval-Segura et al., 2022b; Zhu et al., 2024a). These methods add elaborate and imperceptible perturbations into the training data so that the model cannot learn meaningful information

---

[*]These authors contributed equally to this work.
[†]Corresponding author.

from the data and thereby significantly degrading the test accuracy. A representative method in this domain is the error-minimization poisoning approach (Huang et al., 2020), which employs iterative optimization of a bi-level min-min problem to create poisoning noise. The underlying intuition is that smaller losses trick the model into believing there is nothing to learn from the dataset. Recently, several methods (Fowl et al., 2021; Yuan and Wu, 2021; Sandoval-Segura et al., 2022b; Liu et al., 2024a) have been developed to further enhance the effectiveness of UEs.

However, existing methods primarily rely on empirical heuristics, lacking of convincing explanations. Some studies (Yu et al., 2022; Zhu et al., 2024b) interpret UEs as attempts to create linear shortcuts that yield linear separability, leading models to overwhelmingly depend on spurious features. However, we find that linear classifiers trained on UEs achieve fair generalization (over 30% test accuracy on CIFAR-10), suggesting that interpreting UEs merely as linear shortcuts does not fully explain their much worse generalization in deep neural networks (10% on CIFAR-10, equivalent to random guessing levels). Besides, not all UEs are linearly separable, such as autoregressive poisons (Sandoval-Segura et al., 2022b; 2023). Therefore, the prevailing explanation regarding the linear separability of UEs is incomplete in its applicability. There are still unclear of why UEs are effective, posing significant challenges to further enhancing UEs with better principled approaches.

Unlearnable examples contain elaborately injected poisons and are therefore out-of-distribution. Recently, mutual information (MI) in representation learning, which quantifies the degree of correlation between random variables from two distributions, has gained widespread attention (Oord et al., 2018; Chen et al., 2020). This inspires us to use MI as a surrogate metric to evaluate the unlearnability of UE poisoned datasets. Specifically, we introduce a novel perspective: the reduction of mutual information, to elucidate the underlying mechanism of UEs. We evaluate both test accuracy drop and MI reduction between clean and poisoned features across many UEs, showcasing that effective UEs consistently decrease MI with clean features. Beyond exploring different UEs, we also test the decrease of MI across networks of varying depths. As networks become deeper, MI between features becomes smaller, resulting in test accuracy becomes lower. This consistent relationship between MI reduction and accuracy drop demonstrate our findings.

Based on these analyses, we further enhance the efficacy of UEs by directly decreasing MI between the poisoned and clean distributions in the feature space. However, the complexity of estimating MI poses significant challenges for optimization (Paninski, 2003; McAllester and Stratos, 2020). To tackle this issue, we prove from the perspective of covariance reduction that reducing MI can be achieved by minimizing the conditional covariance of the poisoned data's intra-class features, and we then introduce a novel poisoning method called **Mutual Information Unlearnable Examples (MI-UE)**. Specifically, MI-UE optimizes a mutual information reduction loss that maximizes the cosine similarity among intra-class features for covariance reduction, while minimizing the cosine similarity between inter-class features to prevent class collapse. We conduct extensive experiments to validate that our MI-UE significantly outperforms the previous state-of-the-art UEs in reducing the model's generalization ability. Remarkably, even under defenses such as adversarial training, MI-UE still achieves superior poisoning effects.

## 2   Related Work

**Unlearnable examples.** Privacy issues have received extensive attention in the domain of privacy-preserving machine learning (Shokri and Shmatikov, 2015; Abadi et al., 2016; Shokri et al., 2017), including studies on UEs (Huang et al., 2020). Unlearnable examples are a type of data poisoning that allows an attacker to perturb the training dataset under a small norm restriction. These attacks aim to induce errors during test-time while maintaining the semantic integrity and ensuring the normal usage by legitimate users. A representative method in this field is the error-minimization poisoning strategy (Huang et al., 2020), which employs iterative optimization of a bi-level minimization problem to generate poisoning noise by minimizing the loss. Following the initial work (Huang et al., 2020), additional UE strategies (Yuan and Wu, 2021; Yu et al., 2022; Sandoval-Segura et al., 2022b; Fu et al., 2022; Ren et al., 2022; Liu et al., 2024c;a; Zhu et al., 2024a) have also been proposed. However, existing methods predominantly rely on empirical heuristics and lack a convincing framework to explain the efficacy of UEs, posing significant challenges for advancing UEs in a principled manner.

**Mutual information in machine learning.** Mutual Information (MI) (Shannon, 1948) serves as a metric for quantifying the dependency between two random variables and has been widely applied in

machine learning (Bell and Sejnowski, 1995; Butte and Kohane, 1999; Alemi et al., 2016; Gabrié et al., 2018). Recently, MI maximization in representation learning has gained widespread attention (Oord et al., 2018; Chen et al., 2020). People maximize MI to improve domain adaption (Zhao et al., 2022), and have achieved domain generalization by conducting MI regularization with pre-trained models (Cha et al., 2022). People also have tried to maximize the natural MI and minimize the adversarial MI to enhance adversarial robustness (Zhou et al., 2022). Additionally, MI has been utilized in disentangled representation learning (Chen et al., 2018), cascaded learning (Zhang et al., 2021) and fairness (Zhu et al., 2021). A recent work (Wang et al., 2025) generates UE by minimizing MI between model inputs and outputs. Unlike these approaches, we use MI between clean and poisoned features to establish a generalization upper bound when trained on UEs.

**Mutual information estimation.** Despite the broad application of MI, precise computing or approximating it remains a challenging task (Paninski, 2003; McAllester and Stratos, 2020). Due to exponential growth in sample complexity, traditional approximation methods based on histogram (Pizer et al., 1987; Moddemeijer, 1989), kernel density (Moon et al., 1995), $k$-th nearest neighborhood (Kraskov et al., 2004) struggle in high-dimensional data contexts. Some methods Goldfeld and Greenewald (2021); Goldfeld et al. (2022); Tsur et al. (2023) have used sliced mutual information as a surrogate metric in high-dimensional case. Advanced estimation methods based on deep learning, such as mutual information neural estimator (Belghazi et al., 2018), copula density estimation (Letizia and Tonello, 2022), diffusion-based estimation (Franzese et al., 2023), have been proposed. Chen et al. (2016); Oord et al. (2018); Chen et al. (2020) have introduced lower bound estimation of MI, while another work (Cheng et al., 2020) has proposed an upper bound estimation. However, the complexity of MI estimation still poses significant challenges to optimizing existing approximation methods.

## 3 PRELIMINARY AND MOTIVATION

**Notations.** We denote the data distribution as $\mathcal{D}$, and let $(X, Y) \sim \mathcal{D} = \mathcal{D}_\mathcal{X} \times \mathcal{D}_\mathcal{Y}$ represent the random variables of data instances and their corresponding labels. Consider a classification model $f = h \circ g$, where $g$ is a feature extractor and $h$ is a linear classifier. The feature is $Z = g(X)$.

**Mutual information.** Mutual Information (MI) (Shannon, 1948) serves as a metric for quantifying the dependency between two random variables. Let $X_1$ and $X_2$ be random variables from domains $\mathcal{X}_1$ and $\mathcal{X}_2$, respectively, with marginal probability measures $P_{X_1}$ and $P_{X_2}$, and joint probability measures $P_{X_1, X_2}$. MI measures the discrepancy between $P_{X_1, X_2}$ and $P_{X_1} \times P_{X_2}$:

$$I(X_1, X_2) = \int_{\mathcal{X}_1 \times \mathcal{X}_2} \log \left( \frac{dP_{X_1, X_2}}{d(P_{X_1} \times P_{X_2})} \right) dP_{X_1, X_2}.$$

**Unlearnable examples.** Unlearnable example (UE) is a type of clean-label data poisoning attack, which allows the poison generator to perturb all training data with a small budget (Huang et al., 2020; Feng et al., 2019; Yuan and Wu, 2021; Yu et al., 2022; Sandoval-Segura et al., 2022b). Specifically, the attacker can perturb the training dataset $D = \{(x_i, y_i)\}_{i=1}^N$ into a poisoned version $D' = \{(x_i + \delta_i, y_i)\}_{i=1}^N$, while controlling the $p$-norm of perturbation $\|\delta_i\|_p \leq \epsilon$ to maintain the imperceptibility of poisons. The goal of UEs is to reduce the model's generalization, i.e., degrade the test accuracy, to prevent privacy data from malicious abuse.

**Motivation: why do UEs work?** Yu et al. (2022) has found that UE noises are typically linearly separable, and Zhu et al. (2024b) has further proved that some unlearnable poisoned datasets possess linear separability with high probability when the data dimension is large, and use simple networks to detect potential unlearnable datasets. These studies interpret UEs as attempting to create linear shortcuts for recognition that result in linear separability, leading models to rely overwhelmingly on spurious features for predictions rather than capturing the core features of the images. However, we find that linear classifiers trained on UEs can achieve certain generalization (over 30% test accuracy on CIFAR-10, see Figure 2 and Table 13). Thus, interpreting UEs merely as linear shortcuts does not fully account for why such examples result in much worse generalization in deep neural networks (as low as 10% on CIFAR-10, which is equivalent to random guessing levels, see Table 1). Besides, previous work (Sandoval-Segura et al., 2023) has discovered that not all UEs are linearly separable. For instance, the linear separability of autoregressive poisons (Sandoval-Segura et al., 2022b) is even lower than that of clean images on CIFAR-10. We have also evaluated the linear separability of both unlearnable noise and unlearnable datasets in Appendix G.5. It demonstrates that although

Table 1: The Mutual Information (MI) estimation between clean and poison features on the histogram-based estimator and test accuracy for different UEs on CIFAR-10, along with their gaps from the results for clean data. Compared to random noises, MI for all UEs are significantly reduced.

| Victim ResNet-18 | Clean | Random | EM | AP | NTGA | AR | REM | SEM | GUE | TUE | MI-UE |
|---|---|---|---|---|---|---|---|---|---|---|---|
| Test Acc(%) | 94.45 | 94.11 | 24.17 | 11.21 | 23.11 | 17.41 | 22.94 | 14.78 | 12.04 | 11.25 | 9.95 |
| Acc Gap(%) | - | 0.34 | 70.28 | 83.24 | 71.34 | 77.04 | 71.51 | 79.67 | 82.41 | 83.20 | **84.50** |
| MI | 0.7122 | 0.6747 | 0.6400 | 0.5871 | 0.6126 | 0.5622 | 0.6290 | 0.5747 | 0.5895 | 0.6094 | 0.4969 |
| MI Gap | - | 0.0375 | 0.0722 | 0.1251 | 0.0996 | 0.1500 | 0.0832 | 0.1375 | 0.1227 | 0.1028 | **0.2153** |

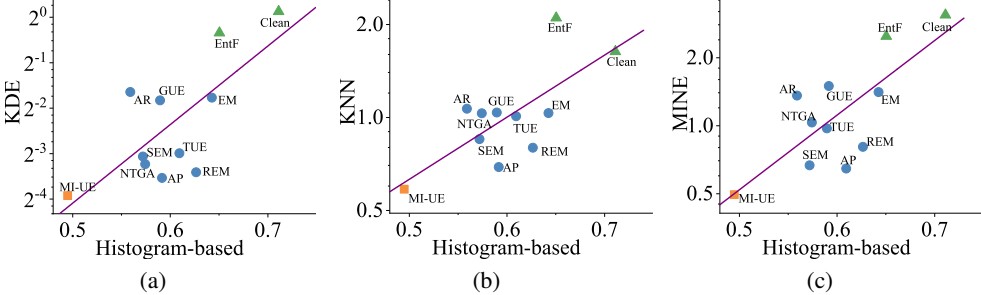

Figure 1: The estimation of MI between clean and unlearnable poisoned features on different MI estimators. **(a):** MI metrics under histogram-based estimator and kernel density estimator (KDE). **(b):** MI metrics under histogram-based estimator and k-NN estimator. **(c):** MI metrics under histogram-based method and mutual information neural estimator (MINE). Green triangles represent clean or ineffective UEs, blue circles mean existing effective UEs, orange square denotes our MI-UE. It demonstrates that although different estimation methods show different quantitative results, the effectiveness of UEs is always positively related with the MI between clean and poisoned features.

many of existing unlearnable examples exhibit linear separability, some methods like AP and AR have poor linear separability close to that of clean data. Thus, the existing explanation based on the linear separability of UEs is incomplete in its applicability. To address these issues and improve the effectiveness of UEs, we propose a novel perspective: *the reduction of mutual information*, to explain the core reason behind the poisoning effect of UEs.

## 4    A NOVEL PERSPECTIVE: MUTUAL INFORMATION REDUCTION

Unlearnable examples, whihc contain elaborately injected poisons, do not belong to the clean data distribution and violate the i.i.d. assumption, posing significant challenges for analyzing their generalization power. Recently, MI in representation learning, which quantifies the degree of correlation between random variables from two distributions, has gained widespread attention (Oord et al., 2018; Chen et al., 2020). This inspires us to use MI as a surrogate metric to evaluate the unlearnability.

**Effective UEs have MI reduction.** Existing studies have empirically constructed UEs from various perspectives, such as deceiving models into perceiving no learnable content (Huang et al., 2020), creating shortcuts (Yu et al., 2022), injecting non-robust features (Fowl et al., 2021), and fooling simple CNNs through autoregressive signals (Sandoval-Segura et al., 2022b). However, our findings indicate that these empirical methods indeed exhibit the reduction of MI between clean features $g(X)$ and poisoned features $g(X')$. We conduct quantitative experiments to measure the changes of MI across different UEs. To facilitate better estimation, we employ sliced mutual information (SMI) (Goldfeld and Greenewald, 2021) to mitigate the challenges on MI estimation for high-dimensional data, and we utilize histogram-based estimator (Moddemeijer, 1989), for one-dimensional MI estimation. Table 1 shows the test accuracy and MI estimation on histogram-based estimator for different UEs, along with their gaps from the results on clean data. We have evaluated the Spearman correlation between Acc gap and MI gap across different unlearnable examples shown in Table 1, the correlation score of

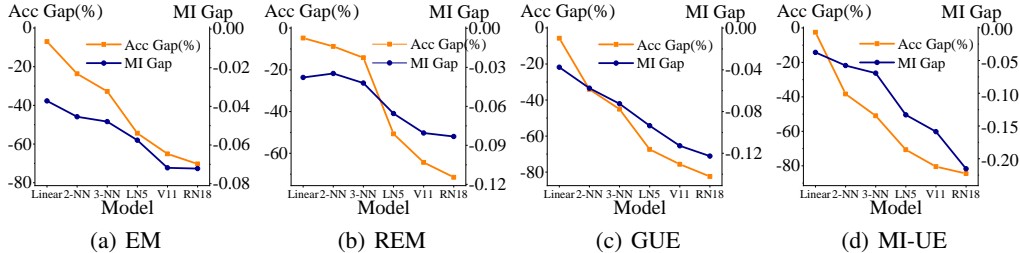

Figure 2: The drop of test accuracy (Acc Gap) and the reduction of MI (MI Gap) of various UEs on different models, including linear, 2-NN, 3-NN, LeNet-5(LN5), VGG-11(V11), ResNet-18(RN18). The results indicate that as the depth and complexity of the models increase, both the drop of test accuracy and the reduction of MI become more pronounced.

0.7818, demonstrating decent positive correlation. The results in Table 1 demonstrate a significant reduction of $I(g(X), g(X'))$ for all UEs compared to those perturbed randomly, supporting our findings that effective UEs have MI reduction.

Relying on a single MI estimator may be unconvincing, as estimating MI is not easy for high-dimensional data (Paninski, 2003; McAllester and Stratos, 2020). To further ensure the confidence of our MI estimation, we also evaluate the estimation of feature MI using other approximation methods, including kernel density estimator (KDE) (Moon et al., 1995), $k$-NN estimator Kraskov et al. (2004), and mutual information neural estimator (MINE) (Belghazi et al., 2018). Details for these estimation methods can be founded in Appendix F.1. The relationship of them compared with Histogram-based estimator is displayed in Figure 1. For every estimator, we first conduct SMI estimator to convert the high dimensional features to one-dimensional ones. Although different estimators show different estimated values, the similar trends unfold for all of these estimators. Specifically, the clean dataset, or ineffective UEs against standard training, ENTF, denoted by green triangles in Figure 1, have shown quite high MI. Existing effective UEs, denoted by blue circles, have demonstrated relatively lower MI. Our proposed MI-UE, by directly minimizing feature's MI, denoted by orange square, has achieved lowest MI for all estimators. Therefore, Figure 1 further increases the confidence of our claim, UEs are caused by MI reduction. Beyond these experimental foundations, we also provide an explanation on relationship between UEs and MI reduction from theoretical views based on some Gaussian assumptions in Appendix A.

**Shallower networks are less affected by UEs.** In addition to exploring different UEs, we conduct experiments to evaluate the decrease of MI across models of varying depths. Similar to ResNet-18 as shown in Table 1, we assess shallower networks including linear model, 2-NN (two-layer neural network), 3-NN (three-layer neural network), LeNet-5 (LeCun et al., 1998), and VGG-11 (Simonyan and Zisserman, 2014). As shown in Figure 2, shallower networks (such as linear model and 2-NN) exhibit smaller reductions of MI (i.e., MI Gap), less drop in test accuracy (i.e., Acc Gap), meaning less effects from UEs. Although shallower networks perform poorly, they are less susceptible to UE attacks. Taking linear network as example, this can be attributed to the feature extractor $g$ being an identity mapping, thus $f = h$, which causes $I(g(X), g(X'))$ to degenerate to $I(X, X')$. Consequently, the small norm constraints on perturbations $X'$ to $X$ severely limit the reduction of MI. In contrast, deeper network's feature extractors $g$ demonstrate superior performance, and due to the error amplification effect (Liao et al., 2018), even norm-constrained perturbations do not limit changes of MI. Therefore, as shown in Figure 2 and Table 13, UEs have a more potent poisoning effect on deeper networks. These results on models of different depths further corroborate the validity of our MI reduction explanation. To further validate the influence of network depth, we also evaluate the test accuracy and MI estimation on ResNet and ViT with different depths in Appendix F.4.

## 5 ACHIEVING UNLEARNABILITY BY MINIMIZING MI

As analyzed in Section 4, the reduction of MI is a primary factor behind the effectiveness of UEs. Therefore, it is natural to consider directly decreasing MI to achieve stronger unlearnability. However, the difficulty in optimizing MI poses significant challenges to the existing optimization methods

(Treves and Panzeri, 1995; Bach and Jordan, 2002; Paninski, 2003). Previous works (McAllester and Stratos, 2020; Belghazi et al., 2018) have noted that all these methods inherently suffer from severe statistical limitations, and have highlighted that optimizing MI using SGD is biased. To address this challenge, we theoretically derive MI from the perspective of covariance reduction. Based on this analysis, we propose a novel unlearnable method, **Mutual Information Unlearnable Examples (MI-UE)**, to generate more effective UEs.

## 5.1 COVARIANCE REDUCTION INDUCES LOW MI

To address the challenges of MI estimation, we present a theorem that introduces a simple assumption on the feature distribution for each class and prove that minimizing the conditional covariance of intra-class features implicitly minimizes MI between distributions.

**Theorem 5.1** (Proof in Appendix B). *Assume that for every $Y \in \mathcal{Y}$, poison distribution $g(X')|Y$ is close to a Gaussian mixture distribution under KL-divergence, i.e., there exists $\mathcal{N}(\mu_Y, \Sigma_Y)$, such that $\mathrm{KL}(\mathcal{N}(\mu_Y, \Sigma_Y)\|p(g(X')|Y)) \leq \epsilon$ for some $0 < \epsilon < 1$. Then, we have:*

$$I(g(X), g(X')) \leq \frac{d}{2}\log(2\pi e) + \frac{1}{2}\mathbb{E}_Y \log(\det \Sigma_Y) + H(g(X')|g(X)) + \mathbb{E}_Y C_Y \sqrt{\epsilon}, \quad (1)$$

*where $C_Y = \sqrt{2} \max_{u \in [m_Y, M_Y]} |\log u| + 1, m_Y = \min p(g(X')|Y), M_Y = \max p(g(X')|Y), I(\cdot, \cdot)$ denotes the mutual information, $H(\cdot|\cdot)$ denotes the conditional entropy, and $d$ is the feature dimension.*

**Remark 5.2.** *Rationality of assumptions in Theorem 5.1 is discussed in Appendix F.5. It is noteworthy that the uncertainty of $H(g(X')|g(X))$ arises solely from two factors: the UE generator $\mathcal{G}: \mathcal{X} \to \mathcal{X}'$ and the training algorithm $\mathcal{A}: \mathcal{X}' \to \mathcal{F}$. Therefore, if both $\mathcal{G}$ and $\mathcal{A}$ are predefined, for example $\mathcal{G}$ as EM and $\mathcal{A}$ as SGD, the third term $H(g(X')|g(X))$ in the equation can be considered as a constant. The term $\frac{d}{2}\log(2\pi e)$ is clearly a constant, thus the critical variable is the covariance $\Sigma_Y$ in the second term.*

As discussed in Section 4, to ensure the effectiveness of UEs, we need to ensure that MI between the poisoned and clean features is minimized. According to Theorem 5.1, this can be achieved by minimizing the conditional covariance of the poisoned features $g(X')|Y$, namely $\Sigma_Y$ when it obeys Gaussian mixture distribution.

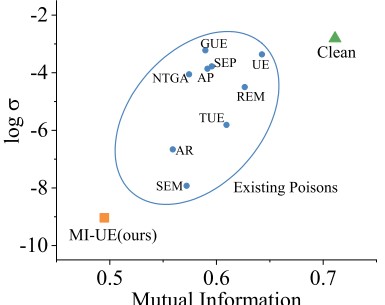

Figure 3: Feature covariance of different unlearnable examples. Results show that all unlearnable methods reduce both MI and covariance, with our MI-UE achieving the lowest values.

## 5.2 MUTUAL INFORMATION UNLEARNABLE EXAMPLES

Based on the theoretical analysis in Section 5.1, we aim to reduce the covariance of $g(X')|Y$. A straightforward approach is to minimize the Euclidean distance between intra-class features $g(X')$. However, normalization techniques such as batch normalization (Ioffe and Szegedy, 2015) and layer normalization (Ba et al., 2016) are commonly employed to enhance the performance of deep models, and the accompanying scaling often leads to the minimization of Euclidean distances between features becoming ineffective. Consequently, we additionally employ cosine similarity loss as a more robust metric (Nguyen and Bai, 2010; Oord et al., 2018). Specifically, we introduce a novel MI reduction loss $\mathcal{L}_{\mathrm{mi}}$ to generate UEs:

$$\mathcal{L}_{\mathrm{mi}}(x+\delta, y; \theta)_j = \log\left(1 + \frac{\sum_{k \in B, y_{b_k} \neq y_{b_j}} \exp(g(x_{b_j} + \delta_{b_j})^T g(x_{b_k} + \delta_{b_k})/\tau)}{\sum_{k \in B, y_{b_k} = y_{b_j}} \exp(g(x_{b_j} + \delta_{b_j})^T g(x_{b_k} + \delta_{b_k})/\tau)}\right)$$
$$+ \zeta \cdot \log(1 + \sum_{k \in B} \|g(x_{b_j} + \delta_{b_j}) - g(x_{b_k} + \delta_{b_k})\|_2), \quad (2)$$

where $\theta$ denotes the model parameters, $B = \{(x_{b_j}, y_{b_j})\}_{j=1}^{N_B}$ is a mini-batch, $j$ is the batch index, $g$ is the feature extractor, $\tau$ is the loss temperature, and $\zeta$ represents the balancing hyperparameter. Within $\mathcal{L}_{\mathrm{mi}}$, we further reduce covariance by maximizing the cosine similarity between intra-class

features. Moreover, we minimize the cosine similarity between inter-class features to prevent class collapse. Inspired by Chen et al. (2020), we employ the exponential operation to simulate the softmax process in cross-entropy loss and the logarithm operation, facilitating better optimization. Beyond optimizing poisons, we also update the shadow model $\theta$ with the cross-entropy loss $\mathcal{L}_{\text{ce}}$, resulting in bi-level optimization by the min-min approach:

$$\min_{\delta} \mathcal{L}_{\text{mi}}(x + \delta, y; \theta^*(\delta)),$$

$$\text{s.t.} \quad \theta^*(\delta) = \arg\min_{\theta} \mathcal{L}_{\text{ce}}(x + \delta, y; \theta). \tag{3}$$

The detailed algorithm, denoted as *Mutual Information Unlearnable Examples (MI-UE)*, is outlined in Algorithm 1. We evaluate the MI and feature covariance of different UEs in Figure 3, compared to clean samples, all UE methods reduce both MI and covariance of poisoned features, with our proposed MI-UE achieving the lowest values of both metrics.

## 6 EXPERIMENTS

### 6.1 EXPERIMENTAL SETUP

**Datasets and models.** In our experiments, we employ three common benchmark datasets: CIFAR-10 (Krizhevsky et al., 2009), CIFAR-100 (Krizhevsky et al., 2009) and ImageNet-subset (Russakovsky et al., 2015) containing the first 100 classes of ImageNet. We evaluated a variety of network architectures, including ResNet-18 (He et al., 2016), ResNet-50 (He et al., 2016), DenseNet-121 (Huang et al., 2017), WRN-34-10 (Zagoruyko and Komodakis, 2016), and ViT-B (Dosovitskiy et al., 2020). Additionally, we utilize shallower networks such as linear network (Linear), two/three-layer fully-connected feed-forward network (2/3-NN) and a classical convolutional network, LeNet-5 (LeCun et al., 1998) for evaluation. More details are provided in Appendix E.

**Baseline methods.** We compare our MI-UE methods with various baseline UEs, including error-minimizing noises (EM) (Huang et al., 2020), strong adversarial poisons (AP) (Fowl et al., 2021), neural tangent attacks (NTGA) (Yuan and Wu, 2021), auto-regressive noises (AR) (Sandoval-Segura et al., 2022b), robust error-minimizing noises (REM) (Fu et al., 2022), stable unlearnable examples (SEM) (Liu et al., 2024c), game-theoretical unleanrable attacks (GUE) (Liu et al., 2024a) and transferable unlearnable examples (TUE) (Ren et al., 2022). For EM, AP, REM, SEM, GUE and TUE, the source model for poison generation is ResNet18. For TUE, the unsupervised backbone is SimCLR. For NTGA, the ensemble model employed is FNN.

**Implementation details.** For UE generation under the $l_{\infty}$ norm, we set the total poison budget at 8/255. In the generation of MI-UE, for CIFAR-10/100, the poisoning epoch is set to 100, at each epoch we conduct 10 steps PGD attack with a step size of 0.2/255. For ImageNet-subset, we set the poisoning epoch to be 50, and the step size of PGD be 0.4/255. The balancing hyperparameter $\zeta$ is set to 0.1 by default. For standard training (ST) evaluation, we use cosine scheduler with an initial learning rate of 0.5, setting the total evaluation epochs at 200. For adversarial training (AT) evaluation, following Fu et al. (2022); Liu et al. (2024c), we set the initial learning rate at 0.1, decaying it by a factor of 10 at epochs 40 and 80, with the total evaluation epochs established at 100. For further details and additional experimental results, please refer to Appendices E and G.

### 6.2 MAIN RESULTS

We evaluate our MI-UE method compared with other baseline methods on three benchmark datasets, CIFAR-10, CIFAR-100 and ImageNet-subset. As demonstrated in Table 2, MI-UE achieves the lowest test accuracy compared to other UEs on the three benchmark datasets, indicating superior poisoning effectiveness. Further assessments are conducted on the transferability across different victim models, including modern deep neural networks such as ResNet, DenseNet, Wide-ResNet, and Vision Transformer, as well as shallower architectures including 2-NN, 3-NN and LeNet-5. As shown in Table 3, our MI-UE consistently results in the lowest test accuracy across all network architectures, establishing it as the most effective form of UEs. Notably, some well-known UEs, such as AP, AR, SEM, and TUE, perform well on modern deep networks but poorly on shallower networks (i.e., 2-NN, 3-NN, and LeNet-5). This disparity suggests a potential sensitivity of these methods to different network architectures. In contrast, our MI-UE exhibits robust transferability across both deep models and shallower architectures. Additional results and analyses are available in Appendix G.3.

Table 2: Quantitative results(%) of baseline methods and our MI-UE for ResNet-18 on three benchmark datasets. Our MI-UE achieves the lowest test accuracy compared to other unlearnable examples, indicating excellent poisoning effectiveness.

| Dataset/Method | Clean | EM | AP | NTGA | REM | SEM | TUE | MI-UE (ours) |
|---|---|---|---|---|---|---|---|---|
| CIFAR-10 | 94.45 | 24.17 | 11.21 | 23.11 | 22.94 | 14.78 | 11.25 | **9.95** |
| CIFAR-100 | 76.65 | 2.09 | 3.73 | 3.08 | 7.52 | 6.29 | 1.34 | **1.17** |
| ImageNet-subset | 80.43 | 1.26 | 9.10 | 8.42 | 13.74 | 4.10 | 4.95 | **1.03** |

Table 3: Quantitative results(%) of baseline methods and our MI-UE on transferability across different models on CIFAR-10. All of unlearnable examples are generated by ResNet-18. Above the dashline represents modern deep networks, while below the dashline represents shallow networks. Our MI-UE consistently results in the lowest test accuracy across all network architectures.

| Model/Method | Clean | EM | AP | NTGA | AR | REM | SEM | GUE | TUE | MI-UE (ours) |
|---|---|---|---|---|---|---|---|---|---|---|
| ResNet-18 | 94.45 | 24.17 | 11.21 | 23.11 | 17.41 | 22.94 | 14.78 | 12.04 | 11.25 | **9.95** |
| ResNet-50 | 95.16 | 23.57 | 11.66 | 19.01 | 15.28 | 23.33 | 13.61 | 12.99 | 10.01 | **9.98** |
| DenseNet-121 | 94.91 | 24.87 | 11.80 | 19.83 | 16.50 | 21.87 | 15.19 | 12.46 | 11.41 | **9.93** |
| WRN34-10 | 96.03 | 24.25 | 11.28 | 21.92 | 14.62 | 21.64 | 13.64 | 13.22 | 12.11 | **10.68** |
| ViT-B | 90.92 | 27.35 | 24.21 | 43.55 | 24.16 | 21.67 | 25.52 | 17.72 | 35.54 | **15.51** |
| LeNet-5 | 80.68 | 26.30 | 31.38 | 44.06 | 73.33 | 29.97 | 22.94 | 13.30 | 28.37 | **10.80** |
| 3-NN | 62.12 | 28.54 | 61.03 | 44.81 | 62.02 | 48.61 | 54.44 | 16.97 | 56.55 | **14.16** |
| 2-NN | 56.15 | 32.50 | 55.78 | 39.34 | 56.75 | 47.37 | 50.79 | 22.08 | 48.75 | **17.82** |

## 6.3 EVALUATION UNDER DEFENSE STRATEGY

**Adversarial training.** For UEs, adversarial training (Madry et al., 2017) is the most straightforward defense mechanism, as UE noises are always constrained by a small norm budget. Adversarial training with the same budget of UEs theoretically minimizes the upper bound of UE generalization risk (Tao et al., 2021). Recent developments have introduced robust UEs specifically targeting adversarial training, such as REM (Fu et al., 2022) and SEM (Liu et al., 2024c). However, these methods are only effective when the adversarial training budget is less than half of the poison budget and fail under larger defense budgets. Another type of UE called ENTF (Wen et al., 2023), that can diminish the efficacy of adversarial training. However, this method is only applicable to large budget of AT and is ineffective for smaller budgets or standard training.

Table 4: Quantitative results(%) of baseline methods and our MI-UE under adversarial training with different budget on CIFAR-10. AT-$i$ means AT with budget $i/255$, ST means standard training. Our MI-UE achieves state-of-the-art performance, particularly achieving an impressive 45.55% at AT-6.

| Method | AT-8 | AT-6 | AT-4 | AT-2 | ST |
|---|---|---|---|---|---|
| Clean | 85.10 | 87.54 | 89.77 | 91.95 | 94.45 |
| EM | 84.57 | 85.42 | 84.29 | 52.81 | 24.17 |
| AP | 82.70 | 85.48 | 88.14 | 22.48 | 11.21 |
| NTGA | 84.22 | 86.27 | 88.36 | 87.87 | 23.11 |
| AR | 84.54 | 87.09 | 89.81 | 92.45 | 17.41 |
| SEM | 85.99 | 86.82 | **29.77** | 19.41 | 14.78 |
| REM | 85.99 | 81.91 | 39.45 | 30.64 | 22.94 |
| ENTF | 75.72 | 76.84 | 77.95 | 81.38 | 91.96 |
| TUE | 84.10 | 86.07 | 89.29 | 91.70 | 11.25 |
| GUE | 84.37 | 86.54 | 71.21 | 17.66 | 12.04 |
| MI-UE | **70.56** | **45.55** | 31.79 | **17.39** | **9.95** |

Our MI-UE method combines the strengths of the above methods, achieving outstanding poisoning effects under both smaller and larger AT budgets. As shown in Table 4, MI-UE achieves the lowest test accuracy under AT settings of 8/255, 6/255, 2/255, and standard training (ST), and it matches the performance of the state-of-the-art robust UE, SEM, at AT-4. Notably, MI-UE demonstrates a significant advantage at AT-8 and AT-6, particularly achieving an impressive 45.55% at AT-6. Thus, our MI-UE achieves state-of-the-art performance under adversarial training defenses. Further details are provided in Appendix G.8.

**Data augmentations.** Standard training often incorporates various data augmentation techniques to improve generalization, such as horizontal flipping and random cropping. To further enhance generalization and prevent overfitting, advanced data augmentation methods have been developed, including Cutout (DeVries and Taylor, 2017), Cutmix (Yun et al., 2019), and Mixup (Zhang et al., 2017). To further evaluate the ability of UEs, we train the victim models under these three aug-

mentation respectively. As shown in the first three columns of Table 5, all existing UEs are highly insensitive to these data augmentations, whereas our MI-UE still achieves the best performance.

**Tailored defense for unlearnable examples.** We evaluate four recent defenses specifically designed for UEs, namely UER (Qin et al., 2023), ISS (Liu et al., 2023), OP (Sandoval-Segura et al., 2023), AVA (Dolatabadi et al., 2023), D-VAE (Yu et al., 2024a) and LE (Jiang et al., 2023). The experimental results are presented in the last six columns of Table 5. UER and OP exhibit inconsistent defensive performance across different UEs. Specifically, they seems work not well for AP, REM, SEM, and

Table 5: Quantitative results(%) of baseline methods and MI-UE under various defense on CIFAR-10. Our MI-UE achieves the lowest test accuracy under the majority of defense methods.

| Defense | Cutout | Cutmix | Mixup | UER | ISS | OP | AVA | D-VAE | LE |
|---|---|---|---|---|---|---|---|---|---|
| Clean | 95.53 | 96.43 | 95.83 | 93.28 | 82.71 | 88.82 | 89.15 | 93.29 | 92.32 |
| EM | 22.90 | 24.08 | 27.22 | 91.41 | 82.78 | 71.70 | 86.62 | 91.42 | 90.93 |
| NTGA | 17.65 | 25.53 | 19.04 | 93.39 | **80.84** | 78.14 | **85.13** | 89.21 | 87.31 |
| AR | 12.84 | 16.20 | 16.24 | 93.32 | 82.79 | 84.69 | 88.38 | 91.11 | 89.75 |
| REM | 26.49 | 24.44 | 18.74 | 69.63 | 82.59 | 29.61 | 86.28 | 86.38 | 90.14 |
| SEM | 14.25 | 15.39 | 15.06 | 70.53 | 81.86 | 23.72 | 87.30 | 88.55 | 88.25 |
| GUE | 13.98 | 20.01 | 12.13 | 85.39 | 83.10 | 86.96 | 86.63 | 90.58 | 84.83 |
| TUE | 11.01 | 10.95 | 11.26 | 92.60 | 82.61 | 85.28 | 88.72 | 91.49 | 85.12 |
| MI-UE | **10.13** | **10.17** | **10.78** | **67.14** | 81.35 | **19.29** | 86.18 | **84.86** | **84.30** |

our MI-UE. UER achieves an accuracy of approximately 70% on these UEs, while OP only manages about 20%. Under these two defenses, D-VAE and LE, our MI-UE maintains the best unlearnability. In contrast, ISS and AVA demonstrate robust performance against existing UEs, with accuracy recovery rates exceeding 80% for all UEs, indicating that bypassing state-of-the-art defenses for UEs remains a challenge, which may be the intrinsic drawback of UEs. Nevertheless, our MI-UE still achieves the best unlearnability in worst-case scenario (86.18% under AVA), the second-best, SEM is 88.55% under D-VAE, other UEs are over 90% in worst-case defenses.

## 6.4 ABLATION STUDIES

**Two terms of the MI-UE loss.** Our MI-UE loss $\mathcal{L}_{\mathrm{mi}}$ includes both similarity term and distance term, so we investigate the effect of them in Table 6. Results show that the similarity term play a more important role for unlearnability, only distance term will significantly degrade the power of MI-UE. Nevertheless, MI-UE with distance term will still increase the unlearnability although it seems a little marginal.

**The strength of balancing hyperparameter.** We conduct the sensitive analysis of the balancing hyperparameter $\zeta$. Results provided in Table 7 demonstrate that the strength be 0.1 or less will slightly increase the effectiveness of MI-UE, while a larger strength like 10 or 100 will significantly destroy the unlearnable power. This phenomenon further reveals the superiority of the similarity measure compared with the simple distance measure.

**Compared with MI-based regularizers.** Inspired by Belghazi et al. (2018), we con-

Table 6: The ablation study on two terms of MI-UE loss. ImageNet-S means ImageNet-subset. Results show that the similarity term play a more important role for unlearnability.

| Method/Dataset | CIFAR-10 | CIFAR-100 | ImageNet-S |
|---|---|---|---|
| MI-UE | 9.95 | 1.17 | 1.03 |
| w/o Distance Term | 10.09 | 2.52 | 1.46 |
| w/o Similarity Term | 51.65 | 26.72 | 23.38 |

Table 7: Sensitive analysis of the balancing hyperparameter strength for MI-UE.

| Strength | 0 | 0.001 | 0.01 | 0.1 | 1 | 10 | 100 |
|---|---|---|---|---|---|---|---|
| Test Acc | 10.09 | 10.03 | 10.08 | 9.95 | 10.31 | 45.90 | 45.47 |

Table 8: The ablation study of MI-UE compared with MI-based regularizers.

| Method | Acc(Acc Gap)(%) | MI(MI Gap) |
|---|---|---|
| UE | 24.17(70.28) | 0.6400(0.0722) |
| UE+MI reg. | 15.62(78.83) | 0.5336(0.1786) |
| AP | 11.21(83.24) | 0.5871(0.1251) |
| AP+MI reg. | 10.01(84.44) | 0.5183(0.1939) |
| MI-UE | **9.95(84.50)** | **0.4969(0.2153)** |

struct an additional MI regularization network to minimize MI. Specifically, we optimize the unlearnable poisons from both the classifier network (ResNet-18) with cross-entropy loss (e.g., UE and AP), and the MINE network (MLP) with MINE loss (a lower bound of MI). All these experiments are conduct on CIFAR-10 dataset. Results in Table 8 demonstrate that under MI loss regularization, both UE and AP show further reduction of MI and drop of test accuracy, further validate our findings: unlearnable examples work because of MI reduction. Moreover, we find that MI regularizations

are still suboptimal compared with our MI-UE, indicating that our algorithm can induce better MI reduction.

**Different training epochs.** As shown in Appendix G.7, generate MI-UE poisons on CIFAR-10/100 requires about 3.6 hours, which is about 1.5x time compared with standard UE's generation like EM. To mitigate the potential computational overheads, we test MI-UE with smaller poisoning epochs in Table 9. Results demonstrate that with half of the generation time (50 epochs), MI-UE still outperforms on CIFAR-10 and achieves the second-best performance on CIFAR-100 compared with existing UEs (Table 2). Furthermore, MI-UE still gains comparable results with other UEs even the poisoning epochs are reduced to 20. Similar results for ImageNet-subset are provided in Appendix G.7. The effectiveness of MI-UE under economic scenarios further demonstrate MI-UE's real-world applications.

Table 9: Quantitative results(%) of MI-UE with different poisoning epochs.

| Epochs/Test Acc(%) | CIFAR-10 | CIFAR-100 |
|---|---|---|
| 100 (default) | 9.95 | 1.17 |
| 50 | 10.25 | 1.66 |
| 20 | 15.39 | 3.23 |

Table 10: Quantitative results(%) of MI-UE with different poisoning budgets.

| Budgets/Test Acc(%) | CIFAR-10 | CIFAR-100 |
|---|---|---|
| 4/255 | 10.49 | 1.16 |
| 6/255 | 10.09 | 1.19 |
| 8/255 (default) | 9.95 | 1.17 |
| 12/255 | 9.83 | 1.17 |
| 16/255 | 9.97 | 1.09 |

**Different poisoning budgets.** We set the poisoning budget $\epsilon = 8/255$ as default to make sure a fair comparison with existing UEs. To validate our MI-UE in broader scenarios, we further generate MI-UE with different poisoning budget from $4/255$ to $16/255$, and evaluate the unlearnability in Table 10. Results demonstrate that even though poisoning budgets across from $4/255$ to $16/255$, MI-UE always results in the unlearnability to a random guess level (i.e., 10% for CIFAR-10, 1% for CIFAR-100).

**Defenses under different JPEG quality in ISS.** We select the JPEG compression in ISS (Liu et al., 2023) with different strengths 6, 8, 10, 12, 15, 20 and 30 as the defense of unlearnable examples, the quantitative results are provided in Table 11. NTGA outperforms on these JPEG quality compressions. Our MI-UE are comparable with REM, achieving the second-best unlearnable performance on average of these JPEG compressions quality. All these unlearnable examples become ineffective against

Table 11: Quantitative results(%) of baseline methods and MI-UE under various JPEG compression quality in ISS defense (Liu et al., 2023). "Average" means the average performance across these compression quality.

| Quality | 6 | 8 | 10 | 12 | 15 | 20 | 30 | Average |
|---|---|---|---|---|---|---|---|---|
| Clean | 78.52 | 81.35 | 83.12 | 83.97 | 85.41 | 86.57 | 88.17 | 83.87 |
| EM | 77.57 | 80.96 | 82.94 | 83.63 | 84.91 | 85.67 | 86.30 | 83.14 |
| NTGA | 76.39 | 78.63 | 79.38 | 79.64 | 81.68 | 81.59 | 81.50 | 79.83 |
| AR | 78.89 | 81.55 | 82.83 | 84.18 | 85.39 | 86.68 | 88.41 | 83.99 |
| REM | 78.04 | 81.30 | 82.20 | 83.23 | 84.50 | 84.60 | 85.24 | 82.73 |
| GUE | 78.27 | 80.69 | 83.00 | 83.97 | 85.39 | 86.46 | 88.06 | 83.69 |
| TUE | 78.48 | 81.83 | 83.23 | 84.13 | 85.12 | 85.88 | 86.38 | 83.58 |
| MI-UE | 77.55 | 80.86 | 81.93 | 83.37 | 84.61 | 85.26 | 84.99 | 82.65 |

JPEG compression, with accuracy recovery to over 80%, indicating that bypassing tailored defense for UEs remains a challenged problem.

## 7 CONCLUSION

In this paper, we introduce a novel perspective for elucidating the mechanisms underlying unlearnable examples: mutual information reduction. We showcase that the harmonious relationship between MI reduction and accuracy drop can be founded in all effective unlearnable examples. Additionally, we derive the mutual information from a covariance reduction standpoint. Based on thi theoretical analysis, we propose a new poisoning method, Mutual Information Unlearnable Examples (MI-UE), which aims to create more effective unlearnable examples by reducing covariance. Extensive experiments consistently demonstrate the superiority of our MI-UE. A limitation of our method is its suboptimal performance under state-of-the-art defenses. However, it is noteworthy that these advanced defenses themselves moderately reduce the accuracy of models trained on clean datasets. We leave further investigation of this aspect to our future work.

ETHICS STATEMENT

This work aims at designing a more effective unlearnable example. Because unlearnable examples are considered as a privacy preserving method to protect data from malicious abuse, we believe our work can bring positive societal impacts in the domain of privacy preserving. For potential negative societal impacts, a malicious entity may seek unlearnable examples to impair normal deep models; defenders can utilize defense strategies on unlearnable examples to avoid this situation.

REPRODUCIBILITY STATEMENT

We provide the implementation details in Section 6.1 and Appendix E to ensure reproducibility. Our codes are available at https://github.com/hala64/mi-ue.

ACKNOWLEDGMENT

This paper is supported by the Strategic Priority Research Program of CAS Grant XDA0480502, NSFC Grants 12288201, 62276149 and 92570001, and the Robotic AI-Scientist Platform of the Chinese Academy of Sciences.

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

# A  THEORETICAL VIEWS OF RELATIONSHIP BETWEEN UES AND MI REDUCTION

**Definition A.1** ($\mathcal{H}$-Divergence, (Ben-David et al., 2010)). *Give the domain $\mathcal{X}$ with $\mathcal{P}$ and $\mathcal{P}'$ probability distributions over $\mathcal{X}$, let $\mathcal{H}$ be a hypothesis class on $\mathcal{X}$ and $I(h)$ be the set of which $h \in \mathcal{H}$ is the characteristic function, i.e., $x \in I(h) \iff h(x) = y$, where $y$ is the label of $x$. The $\mathcal{H}$-Divergence between $\mathcal{P}$ and $\mathcal{P}'$ is defined as*

$$d_{\mathcal{H}}(\mathcal{P}, \mathcal{P}') = 2 \sup_{h \in \mathcal{H}} |\text{Prob}_{\mathcal{P}}[I(h)] - \text{Prob}_{\mathcal{P}'}[I(h)]| \tag{4}$$

**Definition A.2** ($\mathcal{H}\Delta\mathcal{H}$ Space, (Ben-David et al., 2010)). *For a hypothesis space $\mathcal{H}$, the symmetric difference hypothesis space $\mathcal{H}\Delta\mathcal{H}$ is the set of hypotheses that*

$$g \in \mathcal{H}\Delta\mathcal{H} \iff g(x) = h(x) \oplus h'(x) \quad \text{for some } h, h' \in \mathcal{H}, \tag{5}$$

*where $\oplus$ is the XOR function. In other word, $g \in \mathcal{H}\Delta\mathcal{H}$ is the set of disagreement between two hypotheses in $\mathcal{H}$.*

**Theorem A.3** (Proof in Appendix B). *Consider two data distributions $\mathcal{D}$ and $\mathcal{D}'$, with variables $X \sim \mathcal{D}_{\mathcal{X}}$, $X' \sim \mathcal{D}'_{\mathcal{X}}$, and $Y \sim \mathcal{D}_{\mathcal{Y}} = \mathcal{D}'_{\mathcal{Y}}$. Let the classifier be $f = h \circ g$, where $g$ is a feature extractor and $h$ is a linear classifier. Denote $Z = g(X), Z' = g(X')$. For any label $Y$, Assume $Z|Y$ follows a multivariate Gaussian distribution $\mathcal{N}(\mu_{1,Y}, \Sigma_{1,Y})$, $Z'$ relies on $Z$ with $Z'|Y = Z|Y + \mathcal{N}(\mu_{2,Y}, \Sigma_{2,Y})$, and $H(g(\frac{X+X'}{2})) \geq H(\frac{Z+Z'}{2})$. Then we have:*

$$\mathcal{R}_{\mathcal{D}}(f) \leq \mathcal{R}_{\mathcal{D}'}(f) - 4I(Z, Z') + 4H(Y) + \frac{1}{2}d_{\mathcal{H}\Delta\mathcal{H}}(p(Z), p(Z')), \tag{6}$$

*as long as $\det \Sigma_{2,Y} \geq (\frac{2}{\pi e})^d$. Here, $\mathcal{R}_{\mathcal{D}}$ and $\mathcal{R}_{\mathcal{D}'}$ represent the expected population risks over distributions $\mathcal{D}$ and $\mathcal{D}'$, respectively, $I(\cdot, \cdot)$ denotes the mutual information, $H(\cdot)$ represents the entropy, $d_{\mathcal{H}\Delta\mathcal{H}}(\cdot, \cdot)$ is the $\mathcal{H}\Delta\mathcal{H}$-divergence, used to measure the marginal distributions between two logits, $p(\cdot)$ is the probability density function, and $d$ is the feature dimension.*

**Remark A.4.** *We assume that $Z'$ relies on $Z$ because unlearnable examples are crafted from clean dataset. Therefore, their corresponding features $Z'$ and $Z$ should have certain relationship. In our theorem, we assume that they are different by a certain distribution shift.*

**Remark A.5.** *The entropy of $g(\frac{X+X'}{2})$ is not less than $\frac{Z+Z'}{2}$ means that the uncertainty of the former is not less than the latter, this is reasonable when $g$ is a good representation for both $\mathcal{D}$ and $\mathcal{D}'$, the uncertainty of features $Z$ and $Z'$ is relatively low, but the uncertainty of the feature on the mixup data $\frac{X+X'}{2}$ is relatively high.*

**Remark A.6.** *To further make sure the conditional inequality $\det \Sigma_{2,Y} \geq (\frac{2}{\pi e})^d$ holds, we may simply add some random noises for each variable $Z'$, which induce negligible change on performance, to ensure the covariance matrix $\det \Sigma_{2,Y}$ is not nearly singular.*

The last term of Eq. (6) is the $\mathcal{H}\Delta\mathcal{H}$-divergence between the probability density function of clean feature $g(X)$ and poisoned feature $g(X')$. As shown in Definition A.1, $\mathcal{H}\Delta\mathcal{H}$-divergence represents the supremum of absolute probability gap on the true function over hypothesis space $\mathcal{H}\Delta\mathcal{H}$. As shown in Definition A.2, $\mathcal{H}\Delta\mathcal{H}$ space is the symmetric difference space. For instance, with regard to neural network, this space represents the potential different outputs of the neural network on various weights. The $\mathcal{H}\Delta\mathcal{H}$ space reflects the expression power of the hypothesis space $\mathcal{H}$. Because when the $\mathcal{H}$ becomes more complex, there exists more types of $h \in \mathcal{H}$ and $h' \in \mathcal{H}$, such that $h(x)$ and $h'(x)$ represent different functions, making their difference becomes larger. Back to the $\mathcal{H}\Delta\mathcal{H}$-divergence, if the divergence between clean feature $Z = g(X)$ and poisoned feature $Z' = g(X')$ becomes larger, the results of $h \oplus h'(z)$ and $h \oplus h'(z')$ will possibly have much more difference. More simply, we can just regard $\mathcal{H}\Delta\mathcal{H}$ as $\mathcal{M}$, and the function $h \oplus h'$ as the function $m$. If the divergence of $Z$ and $Z'$ is small, the gap between $m(z)$ and $m(z')$ is small too. Conversely, if the divergence becomes larger, the gap of $m(z)$ and $m(z')$ will also become larger, resulting in larger $\mathcal{M}$-divergence. Therefore, the last term somehow represents a divergence between clean and unlearnable features. Furthermore, as the divergence between $Z$ and $Z'$ is getting larger, their MI is expected to become smaller because their relationship becomes weaker. Hence the $\mathcal{H}\Delta\mathcal{H}$-divergence is expected to harmouniouls change with the negative MI.

Theorem A.3 indicates that generalization upper bound under distribution $\mathcal{D}$ increases as the MI between the distributions of features $Z$ and $Z'$ decreases. The condition of $\Sigma_{2,Y}$ is easy to hold (see Remark A.6). Unlearnable examples aim to degrade the generalization on clean distribution $\mathcal{D}$ for classifier $f$ trained on poisoned distribution $\mathcal{D}'$, i.e., increasing $\mathcal{R}_{\mathcal{D}}(f)$. Notably, $\mathcal{R}_{\mathcal{D}'}(f)$ remains minimal since the training set sampled from $\mathcal{D}'$. Furthermore, $H(Y)$ remains constant, as the setting of unlearnable examples does not allow label poisoning. Additionally, the $\mathcal{H}\Delta\mathcal{H}$-divergence somehow represents a divergence between clean and unlearnable features, that can harmoniously change with the negative MI. Therefore, we focus primarily on the MI term $I(Z, Z')$. In this context, Theorem A.3 employs the MI term to establish the generalization upper bound when trained on unlearnable examples.

To ensure the poisoning effect of unlearnable examples, namely higher $\mathcal{R}_{\mathcal{D}}(f)$, the post-poisoning distribution $\mathcal{D}'$ must exhibit a decrease of mutual information $I(Z, Z')$. In the next section, we will demonstrate through experiments that all effective unlearnable examples indeed imply the reduction of MI, validating the correctness and rationality of our proposed theoretical framework.

# B  PROOFS

**Lemma B.1** (Cover (1999)). *If a random variable $X$ obeys the multivariate Gaussian distribution $\mathcal{N}(\mu, \Sigma)$, then the entropy of $X$ is*

$$H(X) = \frac{1}{2} \log \left[ \det\left(2\pi e \Sigma\right) \right]. \tag{7}$$

**Theorem B.2** (Restate of Theorem 5.1). *Assume that for every $Y \in \mathcal{Y}$, poison distribution $Z'|Y$ is close to a Gaussian mixture distribution under KL-divergence, i.e., there exists $\mathcal{N}(\mu_Y, \Sigma_Y)$, such that $\mathrm{KL}(\mathcal{N}(\mu_Y, \Sigma_Y)\|p(Z'|Y)) \leq \epsilon$ for some $0 < \epsilon < 1$. Then, we have:*

$$I(Z, Z') \leq \frac{d}{2} \log(2\pi e) + \frac{1}{2}\mathbb{E}_Y \log(\det \Sigma_Y) + H(Z'|Z) + \mathbb{E}_Y C_Y \sqrt{\epsilon}, \tag{8}$$

*where $C_Y = \sqrt{2} \max\limits_{u \in [m_Y, M_Y]} |\log u| + 1, m_Y = \min\limits_{Z'} p(Z'|Y), M_Y = \max\limits_{Z'} p(Z'|Y), I(\cdot, \cdot)$ denotes the MI, $H(\cdot|\cdot)$ denotes the conditional entropy, and $d$ is the feature dimension.*

*Proof.* Let $P_Y = Z'|Y, Q_Y = \mathcal{N}(\mu_Y, \Sigma_Y)$. As we consider the single-label classification task, for any feature $Z = g(X) \sim \mathcal{Z}$, there exists an unique label $Y$ such that $p(Z|Y) \neq 0$. As UE attack is the clean-label attack, $Z$ and $Z'$ are always assigned with the same label. It holds that

$$
\begin{aligned}
I(Z, Z') &= \int_{z \sim \mathcal{Z}, z' \sim \mathcal{Z}'} p_{Z,Z'}(z, z') \log\left(\frac{p_{Z,Z'}(z, z')}{p_Z(z)p_{Z'}(z')}\right) dz dz' \\
&= \int_{z \sim \mathcal{Z}_y, z' \sim \mathcal{Z}'_y, y \sim \mathcal{D}_y} p_Y(y) p_{Z|Y,Z'|Y}(z|y, z'|y) \log\left(\frac{p_{Z|Y,Z'|Y}(z|y, z'|y)}{p_{Z|Y}(z|y)p_{Z'|Y}(z'|y)}\right) dz dz' dy \\
&= \mathbb{E}_Y I(Z|Y, Z'|Y).
\end{aligned}
$$

Therefore, by the equation between MI and entropy, it has

$$
\begin{aligned}
I(Z, Z') &= \mathbb{E}_{Y \sim \mathcal{D}_y} I(Z|Y, Z'|Y) \tag{9} \\
&= \mathbb{E}_{Y \sim \mathcal{D}_y}\left[H(Z|Y) - H(Z'|Z, Y)\right] \\
&= \mathbb{E}_{Y \sim \mathcal{D}_y}[H(Z'|Y)] - H(Z'|Z) \\
&= \mathbb{E}_{Y \sim \mathcal{D}_y}[H(P_Y)] - H(Z'|Z) \tag{10}
\end{aligned}
$$

As $Q_Y = \mathcal{N}(\mu_Y, \Sigma_Y)$ be the multivariate normal distribution with mean $\mu_Y$ and covariance $\Sigma_Y$, by Lemma B.1, it holds that the entropy of

$$H(Q_Y) = \frac{1}{2} \log\left[(2\pi e)^d \det \Sigma_Y\right] = \frac{d}{2} \log(2\pi e) + \frac{1}{2} \log(\det \Sigma_Y).$$

By Pinsker's inequality, it holds that

$$\mathrm{TV}(P_Y, Q_Y) \leq \sqrt{\frac{1}{2}\mathrm{KL}(Q_Y\|P_Y)} \leq \sqrt{\frac{\epsilon}{2}}.$$

Furthermore, it has

$$
\begin{aligned}
H(P_Y) - H(Q_Y) &= \int \left[ -p_Y(z) \log p_Y(z) + q_Y(z) \log q_Y(z) \right] dz \\
&= \int \left[ q_Y(z) - p_Y(z) \right] \log p_Y(z) dz + \mathrm{KL}(Q_Y \| P_Y) \\
&\leq \max_z |\log p_Y(z)| \int |q_Y(z) - p_Y(z)| dz + \epsilon \\
&\leq 2 \max_{u \in [m_Y, M_Y]} |\log u| \cdot \mathrm{TV}(P_Y, Q_Y) + \epsilon \\
&\leq (\sqrt{2} \max_{u \in [m_Y, M_Y]} |\log u| + 1) \sqrt{\epsilon} \\
&= C_Y \sqrt{\epsilon}.
\end{aligned}
\tag{11}
$$

Then the inequality holds when taking the expectation for every $Y \sim \mathcal{D}_{\mathcal{Y}}$. $\qquad \square$

**Lemma B.3** ((Zhao et al., 2022)). *For two different data distribution $\mathcal{D}$ and $\mathcal{D}'$, $X \sim \mathcal{D}_{\mathcal{X}}, X' \sim \mathcal{D}_{\mathcal{X}'}$. Denote the classifier $f = h \circ g$, in that $g$ is the feature extractor and $h$ is the linear classifier. Then it holds that*

$$
\mathcal{R}_{\mathcal{D}'}(f) \leq \mathcal{R}_{\mathcal{D}}(f) - 4 I_{\frac{\mathcal{D}+\mathcal{D}'}{2}}(Z_1, Z_2) + 4H(Y) + \frac{1}{2} d_{\mathcal{H}\Delta\mathcal{H}}(p(g(X)), p(g(X'))),
\tag{12}
$$

*where $\frac{\mathcal{D}+\mathcal{D}'}{2}$ represents the average mixture of distribution $\mathcal{D}$ and $\mathcal{D}'$, $Z_1 = g(X_1), Z_2 = g(X_2)$, $X_1$ and $X_2$ are sampled from $\frac{\mathcal{D}+\mathcal{D}'}{2}$. $\mathcal{R}_{\mathcal{D}}$ is the expected risk, $d_{\mathcal{H}\Delta\mathcal{H}}(\cdot, \cdot)$ is the $\mathcal{H}\Delta\mathcal{H}$-divergence.*

*Proof of Theorem A.3.* By Lemma B.3, it holds that

$$
\mathcal{R}_{\mathcal{D}}(f) \leq \mathcal{R}_{\mathcal{D}'}(f) - 4I(g(\frac{X+X'}{2}), g(\frac{X+X'}{2})) + 4H(Y) + \frac{1}{2} d_{\mathcal{H}\Delta\mathcal{H}}(P(Z), P(Z')).
$$

Therefore, we only need to prove that

$$
I(Z, Z') \leq I(g(\frac{X+X'}{2}), g(\frac{X+X'}{2}))
$$

to make the inequality in the theorem holds. By the property of mutual information, it holds that

$$
I(Z, Z') = H(Z') - H(Z'|Z)
$$

and

$$
I(g(\frac{X+X'}{2}), g(\frac{X+X'}{2})) = H(g(\frac{X+X'}{2})) \geq H(\frac{Z+Z'}{2})
$$

by the assumption. Hence what we need to prove is

$$
H(Z') - H(Z'|Z) \leq H(\frac{Z+Z'}{2}).
$$

As we are considering the single-label classification task, for any feature $Z = g(X) \sim \mathcal{Z}$, there exists an unique label $Y$ such that $p(Z|Y) \neq 0$, it holds that

$$
\begin{aligned}
H(Z) &= \int_{z \sim \mathcal{Z}} p_Z(z) \log p_Z(z) dz \\
&= \int_{z \sim \mathcal{Z}_y, y \sim \mathcal{D}_{\mathcal{Y}}} p_Y(y) p_{Z|Y}(z|y) \log p_{Z|Y}(z|y) dz dy \\
&= \mathbb{E}_{\mathcal{D}_{\mathcal{Y}}} H(Z|Y),
\end{aligned}
$$

Therefore, it has

$$
H(Z') = \mathbb{E}_Y H(Z'|Y).
$$

Similarly, it holds that

$$
H(Z'|Z) = \mathbb{E}_y H(Z'|Z, Y),
$$

$$H(\frac{Z + Z'}{2}) = \mathbb{E}_Y H(\frac{Z|Y + Z'|Y}{2}).$$

We only need to prove that for all $Y \in \mathcal{D}_\mathcal{Y}$,

$$H(Z'|Y) - H(Z'|Z, Y) \leq H(\frac{Z|Y + Z'|Y}{2}).$$

By Lemma B.1, it has

$$H(Z'|Y) = H(\mathcal{N}(\mu_{1,Y}, \Sigma_{1,Y}) + \mathcal{N}(\mu_{2,Y}, \Sigma_{2,Y})) = \frac{1}{2} \log \left[ (2\pi e)^d \det (\Sigma_{1,Y} + \Sigma_{2,Y}) \right],$$

$$H(Z'|Z, Y) = H(Z'|Z, Y) = H(\mathcal{N}(\mu_{2,Y}, \Sigma_{2,Y})) \frac{1}{2} \log \left[ (2\pi e)^d \det (\Sigma_{2,Y}) \right].$$

$$H(\frac{Z|Y + Z'|Y}{2}) = H(\frac{1}{2} Z|Y + \frac{1}{2} \mathcal{N}(\mu_2, \Sigma_2))$$
$$= \frac{1}{2} \log \left[ (2\pi e)^d \det (\frac{2\Sigma_{1,Y} + \Sigma_{2,Y}}{4}) \right].$$

Thus we need to have the condition provided in the theorem:

$$\det (\Sigma_{1,Y} + \Sigma_{2,Y}) \leq (\frac{\pi e}{2})^d \cdot \det (\Sigma_{2,Y}) \cdot \det (2\Sigma_{1,Y} + \Sigma_{2,Y}).$$

Furthermore, when $\det \Sigma_{2,Y} \geq (\frac{2}{\pi e})^d$, the above condition can be relaxed to

$$\det (\Sigma_{1,Y} + \Sigma_{2,Y}) \leq \det (2\Sigma_{1,Y} + \Sigma_{2,Y}),$$

as covariance matrix is always semi-definite, the eigenvalues of $2\Sigma_{1,Y} + \Sigma_{2,Y}$ will always be non-less than those of $\Sigma_{1,Y} + \Sigma_{2,Y}$, resulting in the above inequality holds. □

## C MORE DISCUSSIONS ON RELATED WORK

**Unlearnable examples.** The safety and trustworthiness of machine learning are becoming increasingly crucial (Szegedy et al., 2013; Carlini and Wagner, 2017; Miao et al., 2022; Cheng et al., 2024; Miao et al., 2024a;b; Cao et al., 2025), with privacy issues having received extensive attention in the domain of trustworthy machine learning (Shokri and Shmatikov, 2015; Abadi et al., 2016; Shokri et al., 2017), including studies on unlearnable examples (Huang et al., 2020; Yu et al., 2022; Sandoval-Segura et al., 2022a; Wen et al., 2023; Li et al., 2023; Lin et al., 2024; Zhu et al., 2024a; Meng et al., 2024; Wu et al., 2025; Yu et al., 2025). Unlearnable examples(UEs) are a type of data poisoning, which allow the attacker to perturb the training dataset under a small norm restriction. These attacks aim to induce errors during test-time while maintaining the semantic integrity and ensuring the normal usage by legitimate users. Many UEs are proposed in recent years, include generating bi-level error-minimizing noises (Huang et al., 2020), using non-robust features for strong adversarial poisons (Fowl et al., 2021), attacking by neural tangent kernels (Yuan and Wu, 2021), inducing linear shortcuts (Yu et al., 2022), fooling convolutional networks by autoregressive signals (Sandoval-Segura et al., 2022b), injecting class-wise separability discriminant (Ren et al., 2022), solving Stackelberg equilibrium of UE game (Liu et al., 2024a). Furthermore, UEs have been extended to adversarial training (Fu et al., 2022; Wen et al., 2023; Liu et al., 2024c), self-supervised learning (He et al., 2022; Ren et al., 2022), unsupervised learning (Zhang et al., 2023), natural language processing (Li et al., 2023), multimodal contrastive learning (Liu et al., 2024b), securing medical data (Sun et al., 2024), 3D point clouds (Zhu et al., 2024a; Wang et al., 2024a). Recent works try to detect UEs by linear separability (Zhu et al., 2024b), iterative filtering (Yu et al., 2024b), edge pixel detector (Li et al., 2025), and dataset watermarking (Zhu et al., 2025). Defense methods for UEs have also been developed recently, including adversarial training (Tao et al., 2021), shortcuts squeezing (Liu et al., 2023), orthogonal projection (Sandoval-Segura et al., 2023), stronger data augmentations (Qin et al., 2023; Zhu et al., 2024b), diffusion purification (Jiang et al., 2023; Dolatabadi et al., 2023; Yu et al., 2024a). Discussions and measures of UEs across multiple tasks have also been studied recently (Ye et al., 2025).

## D   ALGORITHM

Our generating process of Mutual Information Unlearnable Examples (MI-UE) has provided in Algorithm 1. It is shown that our MI-UE adopts the bi-level min-min optimization, at each epoch, we first update the source model to mimic the training process of victim models who use UEs as their training set. Then we update our MI-UE poisons using $\mathcal{L}_{mi}$ by PGD attacks. Source model optimization tries to make $\mathcal{R}_{\mathcal{D}'}(f)$ minimal since $f$ in trained on the UE dataset sampled from $\mathcal{D}'$. UE poison optimization tries to minimize the feature MI between $g(X)$ and $g(X')$ to increase the generalization upper bound of clean data distribution $\mathcal{R}_{\mathcal{D}}(f)$ to enhance the unlearnability.

---

**Algorithm 1** Mutual Information Unlearnable Examples (MI-UE)

---

**Input:** A training dataset $D = \{(x_i, y_i)\}_{i=1}^N$. Total epoch $T$. Batch size $N_B$. MI reduction loss $\mathcal{L}_{mi}$. Model optimization parameters $\alpha_\theta$ and $T_\theta$. UEs optimization parameters $\alpha_\delta$, $T_\delta$ and $T_a$, poison budget $\epsilon$.
**Output:** Poisons $\{\delta_i\}_{i=1}^N$
**Initialize:** $\delta_i \leftarrow 0, i = 1, 2, \cdots, N$
**for** $t = 1, \cdots, T$ **do**
    **for** $t_\theta = 1, \cdots, T_\theta$ **do**                                    ▷ Source model optimization
        Sample a mini batch $B = \{(x_{b_j}, y_{b_j})\}_{j=1}^{N_B}$.
        $\theta \leftarrow \theta - \alpha_\theta \cdot \nabla_\theta \mathbb{E}_{(x_{b_j}, y_{b_j}) \in B} \left[ \mathcal{L}_{ce}(x_{b_j} + \delta_{b_j}, y_{b_j}; \theta) \right]$
    **for** $t_\theta = 1, \cdots, T_\delta$ **do**                        ▷ Unlearnable examples optimization
        Sample a mini batch $B = \{(x_{b_j}, y_{b_j})\}_{j=1}^{N_B}$.
        **for** $t_a = 1, \cdots, T_a$ **do**                    ▷ PGD attacks
            $\delta_{b_j} \leftarrow \delta_{b_j} - \alpha_\delta \cdot \nabla_{\delta_{b_j}} \mathbb{E}_{(x_{b_j}, y_{b_j}) \in B} \left[ \mathcal{L}_{mi}(x_{b_j} + \delta_{b_j}, y_{b_j}; \theta) \right]$
            $\delta_{b_j} \leftarrow \Pi(\delta_{b_j}, -\epsilon, \epsilon)$                    ▷ Clip poisons to $\epsilon$-ball

---

## E   EXPERIMENTAL DETAILS

### E.1   DATASETS

**CIFAR-10/100.** CIFAR-10 and CIFAR-100 (Krizhevsky et al., 2009) contain 50000 training images and 10000 test images, with 10 and 100 classes respectively. The image size is $32 \times 32$ with 3 color channels.

**ImageNet-subset.** ImageNet-subset contains the first 100 classes of the ImageNet-1k dataset from ImageNet ILSVRC (Russakovsky et al., 2015). It has 130000 images as the training set, and 5000 images as the test set. The images have been processed to $224 \times 224$ size with 3 color channels.

### E.2   MODELS

**ResNet.** We use the Deep Residual Network (He et al., 2016) with 18 layers and 50 layers respectively, denoted ResNet-18 and ResNet-50.

**DenseNet.** We employ the Densely Connected Convolutional Networks (Huang et al., 2017) with 121 layers, denoted as DesNet-121.

**Wide ResNet.** We conduct the Wide Residual Networks (Zagoruyko and Komodakis, 2016) with depth be 34 and width factor be 10, denoted as WRN34-10.

**ViT.** We use the vision transformer proposed by (Dosovitskiy et al., 2020) with their base configuration, denoted as ViT-B. For CIFAR-10/100, we change the patch size from 16 to 4.

### E.3   DATA AUGMENTATION

For CIFAR-10/100, we include standard data augmentations for both UE generation and victim model evaluation. We use the Random Crop with the size be 32 and padding be 4, and the Random Horizontal Flip with probability be 0.5. For test set we do not conduct any data augmentation.

For ImageNet-subset, as each raw image is with different size, for both UE generation and victim model evaluation, in the training set, we directly resize the image to the size of $224 \times 224$, follow with the Random Horizontal Flip with probability be 0.5. In the evaluation phase, for the test set, we first resize the image to the size of $256 \times 256$, then we conduct the Center Crop to the size of $224 \times 224$.

### E.4 TRAINING DETAILS

For UE generation, we set the total epoch $T$ be 100 for CIFAR-10/100, and 50 for ImageNet-subset. The batch size $N_B$ is set to be 512. For source model optimization, we use the cross-entropy loss and the mini-batch SGD optimizer, with initial learning rate be 0.5, momentum be 0.9 and weight decay be $1 \times 10^{-4}$. We also conduct the cosine annealing schedule to adjust learning rate at each epoch, with the final minimum learning rate be $1 \times 10^{-6}$. For poison optimization, we use our proposed MI reduction loss with PGD attacks. The temperature of similarity term is set to be 0.1, the strength of distance item is set to be 0.1 by default. We use 10 steps of PGD attack, with each step size be 0.4/255 for ImageNet-subset, and use 10 steps with each step size be 0.2/255 for CIFAR-10/100, and the overall poison budget is set to be 8/255 under $l_\infty$ norm.

For victim model evaluation, we set the training epoch be 200, the batch size be 128 and use SGD optimizer with initial learning rate be 0.5, momentum be 0.9 and weight decay be $1 \times 10^{-4}$, while adjusting learning rate by cosine annealing schedule with the final minimum learning rate be $1 \times 10^{-6}$. For adversarial training on victim models, similar with (Fu et al., 2022) and (Liu et al., 2024c), we change the training epoch to 100, and use the multi step learning rate scheduler, with the learning rate decay by 0.1 at the 40-th and 80-th epoch.

## F MORE DISCUSSION AND EXPERIMENTS ON MUTUAL INFORMATION

In the domain of UEs, any feature $Z \sim \mathcal{Z}$ will have a unique label $Y$ such that the conditional probability $p(Z|Y) \neq 0$ because we consider the single-label classification task. In other words, the feature distribution $\mathcal{Z}$ is class-wise separable. We assume that the data distribution is balanced, i.e., the label distribution is uniform, $p_Y(y) = \frac{1}{C}$, where $C$ is the number of classes. The feature of $X$, under label $Y$, i.e., $g(X)|Y$ satisfies certain distribution. This is reasonable since people usually use class-conditional distribution $p_{X|Y}(x|y)$ to depict data distribution for classification tasks (Bishop, 2006). Therefore, a well-trained feature extractor $g$ will divide data from different labels into different feature region, then a linear classifier $h$ can easily grasp features of each class to their corresponding logits.

In this case, as shown in Equation (9) in the proof of Theorem 5.1, the feature mutual information $I(g(X), g(X'))$ can be divided into the class-conditional one, i.e., $I(g(X), g(X')) = \mathbb{E}_Y I(g(X)|Y, g(X')|Y)$. Therefore, we evaluate the feature MI for data in each class, and then take the average of them as the final estimation of $I(g(X), g(X'))$. For class-wise feature MI estimation, as we assume that they satisfy certain distribution, we can estimate them by various density estimators, details are provided in the following section.

### F.1 DETAILS ON ESTIMATION OF MUTUAL INFORMATION

**Histogram-based estimator.** Histogram is a traditional method for density estimation (Pizer et al., 1987; Moddemeijer, 1989), which is to partition the data set into several bins and use the count of bins as the estimation of density. Precisely, denote the (one-dimensional) data lies in $[0, 1]$, the space is divided into $b$ disjoint bins, written as $B_i = [(i-1)/b, i/b], i = 1, 2, \cdots, b$. For $x \in B_i$, the density $p(x)$ is estimated as

$$p(x) = \frac{b}{n} \sum_{j=1}^{n} \mathbb{I}(X_j \in B_i), \tag{13}$$

where $n$ is the number of samples. In our paper, we set the bin to be 100 by default. The MI is then estimated by corresponding marginal distribution $p(x), p(y)$ and joint distribution $p(x, y)$.

**Kernel density estimator.** Kernel density estimator (Moon et al., 1995) uses the KDE function

$$p(x) = \frac{1}{nh} \sum_{i=1}^{n} K(\frac{x - X_i}{h}) \tag{14}$$

as the estimation of one-dimensional data $x$, where $K(\cdot)$ is the kernel function and $h$ is the bandwidth. In our paper, we use the Gaussian kernel and Silverman rule-of-thumb bandwidth estimator (Silverman, 2018) for the bandwidth selection. The MI is then estimated by corresponding marginal distribution $p(x), p(y)$ and joint distribution $p(x, y)$.

**$k$-NN estimator.** $k$-nearest neighbor (Kozachenko and Leonenko, 1987; Kraskov et al., 2004) is another estimator for entropy and MI. They first compute the $k$-th neighbor distance $\rho_{i,k}$ on joint distribution for each data point $Z_i = (X_i, Y_i), i = 1, \cdots, N$, and then compute the number of neighbors for each $X_i$ and $Y_i$ respectively, denoted as $N_{X_i}$ and $N_{Y_i}$. In detail, it holds that

$$N_{X_i} = \big| \{x_j; d(x_i, x_j) < \rho_{i,k}, i \neq j\} \big|.$$

After that the mutual information $I(X, Y)$ is estimated by

$$\hat{I}(X, Y) = \psi(k) + \psi(N) - \frac{1}{N} \sum_{i=1}^{n} [\phi(N_{X_i} + 1) + \phi(N_{Y_i} + 1)], \tag{15}$$

where $\psi(x)$ is the digamma function defined as $\psi(x) = \frac{d\Gamma(x)}{\Gamma(x)dx}$, $\Gamma(\cdot)$ is the Gamma function. In this paper, we set $k = 3$ by default.

**Mutual information neural estimator (MINE).** MINE (Belghazi et al., 2018) proposed to use neural network as the estimator of MI, by approximating a variational lower bound of MI under Donsker-Varadhan representation (Donsker and Varadhan, 1983):

$$\hat{I}(X, Y) = \sup_{\theta} \mathbb{E}_{(X,Y) \sim p(x,y)}[T_\theta(X, Y)] - \log \mathbb{E}_{X \sim p(x), Y \sim p(y)}[e^{T_\theta(X,Y)}], \tag{16}$$

where $T_\theta$ is the function parameterized by neural network $\theta$ which can be trained on data points $X_i$ and $Y_i$. In this paper, we set the batch size be 1000 and the training iteration be 500 when training the MINE.

**Sliced mutual information.** Sliced mutual information (SMI) (Goldfeld and Greenewald, 2021) is a surrogate measure of MI for high dimensional data, which is defined as

$$SI(X, Y) = \frac{1}{S_{d_x-1} S_{d_y-1}} \oint_{\mathbb{S}^{d_x-1}} \oint_{\mathbb{S}^{d_y-1}} I(\theta^T X, \phi^T Y) d\theta d\phi, \tag{17}$$

in that $\mathbb{S}^{d-1}$ is the $d$-dimensional unit sphere, $S_{d-1}$ is the surface area of $\mathbb{S}^{d-1}$. In practice, as shown on Algorithm 1 in (Goldfeld and Greenewald, 2021), SMI can be estimated by randomly sampled coefficient on $d_x$ and $d_y$-dimensional unit sphere, then compute the one-dimensional MI based on other estimation methods. In this paper, we set the slices number $m$ be 2000 by default.

### F.2 MORE RESULTS ON MI OF UEs UNDER VARIOUS DEFENSES

In this section, we provide the quantitative results of MI and accuracy on several defense methods, including AT, Cutout, UER and ISS, for various UEs on CIFAR-10 dataset.

Results shown in Table 12 that the Acc Gap have quite strong correlation with the MI gap. For instance, the traditional data augmentation technique, Cutout, which are ineffective, show similar MI gaps compared with standard training, that all of existing UEs obtain MI reduction, while our MI-UE achieves the largest. For adversarial training, all of existing UEs become ineffective as well as approaching MI gap compared with random noises, while our MI-UE achieves decent unlearnability as well as further MI drop. For tailored defenses (UER and ISS), similar trends also display, showcasing that MI gaps successfully reflect the unlearnability across different attacks and defense mechanisms.

Table 12: Test accuracy and MI estimation on various UEs compared with clean CIFAR-10 dataset under defense mechanism including AT, Cutout, UER and ISS.

| Defense | Method | Clean | Random | EM | AP | NTGA | AR | REM | SEM | GUE | TUE | MI-UE |
|---------|--------|-------|--------|-----|-----|------|-----|------|------|------|------|-------|
| AT | Acc(%) | 85.10 | 85.07 | 84.57 | 82.70 | 84.22 | 84.54 | 85.99 | 85.99 | 84.37 | 84.10 | 70.56 |
| | Acc Gap(%) | - | 0.03 | 0.53 | 2.40 | 0.88 | 0.56 | -0.79 | -0.79 | 0.73 | 1.00 | **14.54** |
| | MI | 0.7040 | 0.6516 | 0.6309 | 0.6348 | 0.6330 | 0.6377 | 0.6396 | 0.6323 | 0.6388 | 0.6400 | 0.6125 |
| | MI Gap | - | 0.0524 | 0.0731 | 0.0692 | 0.0710 | 0.0663 | 0.0644 | 0.0717 | 0.0652 | 0.0640 | **0.0915** |
| Cutout | Acc(%) | 95.53 | 95.57 | 22.90 | 11.30 | 17.65 | 12.84 | 26.49 | 14.25 | 13.98 | 11.01 | 10.13 |
| | Acc Gap(%) | - | -0.04 | 72.63 | 84.23 | 77.88 | 82.69 | 69.04 | 81.28 | 81.55 | 84.52 | **85.40** |
| | MI | 0.7157 | 0.6739 | 0.6385 | 0.5992 | 0.5883 | 0.5820 | 0.6326 | 0.5753 | 0.5914 | 0.5937 | 0.4982 |
| | MI Gap | - | 0.0418 | 0.0772 | 0.1165 | 0.1274 | 0.1337 | 0.0831 | 0.1404 | 0.1243 | 0.1220 | **0.2175** |
| UER | Acc(%) | 93.28 | 93.30 | 91.41 | 70.65 | 93.39 | 93.32 | 69.63 | 70.53 | 85.39 | 92.60 | 67.14 |
| | Acc Gap(%) | - | -0.02 | 1.87 | 22.63 | -0.11 | -0.04 | 23.65 | 22.75 | 7.89 | 0.68 | **25.14** |
| | MI | 0.7127 | 0.6725 | 0.6688 | 0.6323 | 0.6604 | 0.6701 | 0.6436 | 0.6260 | 0.6593 | 0.6526 | 0.5819 |
| | MI Gap | - | 0.0402 | 0.0439 | 0.0804 | 0.0523 | 0.0426 | 0.0691 | 0.0867 | 0.0534 | 0.0601 | **0.1308** |
| ISS | Acc(%) | 82.71 | 82.66 | 82.78 | 82.50 | 80.84 | 82.79 | 82.59 | 81.86 | 83.10 | 82.61 | 81.35 |
| | Acc Gap(%) | - | 0.05 | -0.07 | 0.21 | **1.87** | -0.08 | 0.12 | 0.85 | -0.39 | 0.10 | 1.36 |
| | MI | 0.7059 | 0.6709 | 0.6680 | 0.6543 | 0.6552 | 0.6598 | 0.6722 | 0.6532 | 0.6688 | 0.6683 | 0.6327 |
| | MI Gap | - | 0.0350 | 0.0379 | 0.0516 | 0.0507 | 0.0461 | 0.0337 | 0.0527 | 0.0371 | 0.0376 | **0.0732** |

## F.3 MORE RESULTS ON MI OF UEs UNDER DIFFERENT NETWORK STRUCTURES

In this section, we provide detailed results of the test accuracy and MI of EM, REM, GUE and MI-UE unlearnable examples compared with clean CIFAR-10 dataset on different networks, including Linear Models, 2-NN, 3-NN, LeNet-5, VGG-11 and ResNet-18. The results provided in Table 13 further show that shallower networks (such as Linear and 2-NN) exhibit relatively higher test accuracy, with lesser declines in test accuracy, and smaller reductions of MI. When the networks become deeper and more complex, the test accuracy on UEs become smaller, with the drop of test accuracy as well as the reduction of MI become greater.

Table 13: Test accuracy and MI estimation between EM, REM, GUE and MI-UE unlearnable examples compared with clean CIFAR-10 dataset under victim linear classifiers (Linear), two-layer neural network (2-NN), three-layer neural network (3-NN), LeNet-5, VGG-11 and ResNet18.

| Method | Model | Linear | 2-NN | 3-NN | LeNet-5 | VGG-11 | ResNet-18 |
|--------|-------|--------|------|------|---------|--------|-----------|
| Clean | Acc(%) | 39.13 | 56.15 | 62.41 | 80.68 | 91.44 | 94.45 |
| | MI | 0.6682 | 0.6850 | 0.7197 | 0.7222 | 0.7583 | 0.7122 |
| EM | Acc(%) | 32.09 | 32.5 | 29.63 | 26.30 | 26.34 | 24.17 |
| | Acc Gap(%) | -7.04 | -23.65 | -32.78 | -54.38 | -65.10 | -70.28 |
| | MI | 0.6310 | 0.6396 | 0.6718 | 0.6647 | 0.6865 | 0.6400 |
| | MI Gap | -0.0372 | -0.0454 | -0.0479 | -0.0575 | -0.0718 | -0.0722 |
| REM | Acc(%) | 34.37 | 47.37 | 48.22 | 29.97 | 27.09 | 22.94 |
| | Acc Gap(%) | -4.76 | -8.78 | -14.19 | -50.71 | -64.35 | -71.51 |
| | MI | 0.6304 | 0.6503 | 0.6776 | 0.6567 | 0.6778 | 0.6290 |
| | MI Gap | -0.0378 | -0.0347 | -0.0421 | -0.0655 | -0.0805 | -0.0832 |
| GUE | Acc(%) | 33.41 | 22.08 | 17.33 | 13.3 | 15.8 | 12.04 |
| | Acc Gap(%) | -5.72 | -34.07 | -45.08 | -67.38 | -75.64 | -82.41 |
| | MI | 0.6307 | 0.6274 | 0.6472 | 0.6286 | 0.6455 | 0.5895 |
| | MI Gap | -0.0375 | -0.0576 | -0.0725 | -0.0936 | -0.1128 | -0.1227 |
| MI-UE | Acc(%) | 36.60 | 17.82 | 11.43 | 10.01 | 10.98 | 9.95 |
| | Acc Gap(%) | -2.53 | -38.33 | -50.98 | -70.67 | -80.46 | -84.50 |
| | MI | 0.6305 | 0.6273 | 0.6504 | 0.5892 | 0.5997 | 0.4969 |
| | MI Gap | -0.0377 | -0.0577 | -0.0693 | -0.1330 | -0.1586 | -0.2153 |

## F.4 MORE RESULTS ON MI OF UEs UNDER DIFFERENT NETWORK DEPTHS

To further validate the influence of network depth, we evaluate the test accuracy and MI estimation between UE, REM, GUE and MI-UE unlearnable examples compared with clean CIFAR-10 dataset

on ResNet-34, ResNet-50, ResNet-101 and ResNet-152. (parentheses mean the gap of given method compared with clean data.)

Table 14: Test accuracy and MI estimation between EM, REM, GUE and MI-UE unlearnable examples compared with clean CIFAR-10 dataset under ResNet-18, ResNet-34, ResNet-50, ResNet-101 and ResNet-152.

| Method | Model | ResNet-18 | ResNet-34 | ResNet-50 | ResNet-101 | ResNet-152 |
|---|---|---|---|---|---|---|
| Clean | Acc(%) | 94.45 | 94.63 | 95.16 | 95.43 | 95.55 |
| | MI | 0.7122 | 0.7067 | 0.7078 | 0.7022 | 0.7035 |
| EM | Acc(%) | 24.17 | 23.96 | 23.57 | 23.75 | 23.32 |
| | Acc Gap(%) | -70.28 | -70.67 | -71.59 | -71.68 | -72.23 |
| | MI | 0.6400 | 0.6318 | 0.6285 | 0.614 | 0.6149 |
| | MI Gap | -0.0722 | -0.0749 | -0.0793 | -0.0876 | -0.0886 |
| REM | Acc(%) | 22.94 | 22.99 | 23.33 | 21.12 | 20.75 |
| | Acc Gap(%) | -71.51 | -71.64 | -71.83 | -74.31 | -74.80 |
| | MI | 0.6290 | 0.6228 | 0.6169 | 0.6100 | 0.6072 |
| | MI Gap | -0.0832 | -0.0839 | -0.0909 | -0.0922 | -0.0963 |
| GUE | Acc(%) | 12.04 | 12.02 | 12.99 | 12.93 | 11.85 |
| | Acc Gap(%) | -82.41 | -82.61 | -82.17 | -82.50 | -83.70 |
| | MI | 0.5895 | 0.5787 | 0.5704 | 0.5623 | 0.5610 |
| | MI Gap | -0.1227 | -0.1280 | -0.1374 | -0.1399 | -0.1425 |
| MI-UE | Acc(%) | 9.95 | 9.97 | 9.98 | 9.99 | 9.90 |
| | Acc Gap(%) | -84.50 | -84.66 | -85.18 | -85.44 | -85.65 |
| | MI | 0.4969 | 0.4857 | 0.4802 | 0.4621 | 0.4338 |
| | MI Gap | -0.2153 | -0.2210 | -0.2276 | -0.2401 | -0.2697 |

We also evaluate Acc gap and MI gap with UE, REM, GUE, MI-UE unlearnable examples on CIFAR-10 dataset under three types of vision transformer proposed by Dosovitskiy et al. (2020), namely, ViT-Base (depth=12), ViT-Large (depth=24) and ViT-Huge (depth=32).

Table 15: Test accuracy and MI estimation between EM, REM, GUE and MI-UE unlearnable examples compared with clean CIFAR-10 dataset under ViT-Base, ViT-Large and ViT-Huge.

| Method | Model | ViT-Base | ViT-Large | ViT-Huge |
|---|---|---|---|---|
| Clean | Acc(%) | 90.92 | 91.35 | 91.93 |
| | MI | 0.7285 | 0.7297 | 0.7274 |
| EM | Acc(%) | 27.35 | 26.26 | 25.87 |
| | Acc Gap(%) | -63.57 | -65.09 | -66.06 |
| | MI | 0.6531 | 0.6449 | 0.6370 |
| | MI Gap | 0.0754 | -0.0848 | -0.0904 |
| REM | Acc(%) | 21.67 | 21.37 | 20.49 |
| | Acc Gap(%) | -69.25 | -69.98 | -71.44 |
| | MI | 0.6427 | 0.6241 | 0.6183 |
| | MI Gap | -0.0858 | -0.1056 | -0.1091 |
| GUE | Acc(%) | 17.72 | 17.03 | 16.58 |
| | Acc Gap(%) | -73.20 | -74.32 | -75.35) |
| | MI | 0.5988 | 0.5923 | 0.5806 |
| | MI Gap | -0.1297 | -0.1374 | -0.1468 |
| MI-UE | Acc(%) | 15.51 | 14.43 | 13.78 |
| | Acc Gap(%) | -75.41 | -76.92 | -78.15) |
| | MI | 0.5332 | 0.5296 | 0.5180 |
| | MI Gap | -0.1953 | -0.2001 | -0.2094 |

Results in Tables 14 and 15 show that, as the network become deeper, the drop of test accuracy (Acc Gap) and the MI reduction (MI Gap) always become bigger harmoniously, further validate the relationship between unlearnability and MI reduction. Our MI-UE achieves both greatest unlearnability (lowest Acc) as well as best MI reduction (Highest MI Gap).

Table 16: The unlearnable class accuracy and other classes accuracy of various one-class UEs on CIFAR-10.

| Test Acc(%) | Unlearnable Class | Other Classes |
|---|---|---|
| Clean | 95.9 | 94.3 |
| EM | 2.2 | 95.4 |
| AP | 0.2 | 95.3 |
| NTGA | 4.0 | 95.1 |
| AR | 0.7 | 94.9 |
| SEM | 0.3 | 94.9 |
| REM | 0.5 | 95.6 |
| TUE | 0.1 | 95.0 |
| GUE | 0.3 | 95.1 |
| MI-UE | **0.0** | 95.1 |

Table 17: The unlearnable class accuracy and other classes accuracy of various one-class UEs on CIFAR-100.

| Test Acc(%) | Unlearnable Class | Other Classes |
|---|---|---|
| Clean | 88 | 76.6 |
| EM | 8 | 76.7 |
| AP | 3 | 76.2 |
| NTGA | 26 | 77.0 |
| AR | 90 | 77.1 |
| SEM | 18 | 75.9 |
| REM | 12 | 76.8 |
| TUE | 7 | 76.5 |
| GUE | 86 | 77.4 |
| MI-UE | **1** | 77.1 |

### F.5 RATIONALITY OF ASSUMPTION FOR GAUSSIAN MIXTURE DISTRIBUTION

From an empirical perspective, modern neural networks in feature space, after normalization operations such as BatchNorm/LayerNorm, often present a nearly isotropic distribution, which has a similar covariance structure to the Gaussian distribution (Daneshmand et al., 2021). Even if the actual distribution deviates from Gaussian (with $\epsilon$ KL divergence gap), the upper bound given by Theorem 5.1 is still a function of $\Sigma_Y$, and our MI-UE actually optimizes by maximizing the cosine similarity of similar features to compress the intra-class covariance. Therefore, MI-UE doesn't rely on strict Gaussianity, as long as the inter-class/intra-class covariance is controllable, it can play a role in MI reduction and achieving unlearnability.

## G    ADDITIONAL EXPERIMENTS

### G.1    ONE-CLASS UNLEARNABLE EXAMPLES

In real-world scenarios, the privacy protector sometimes only have access to their own class of data rather than the whole dataset, for instance, when people upload their selfies into the social media, they can only modify their own images. Therefore, we also investigate various UEs when only one class of the whole dataset is perturbed. More precisely, we only add perturbations to the class 0 of CIFAR-10 (i.e., the class "plane"), and evaluate the "plane" class accuracy as well as accuracy of other classes. Results provided in Table 16 reveal that, existing one-class UEs can make the perturbed class be unlearnable, while keeping the accuracy of other classes remain or become even higher. Furthermore, our MI-UE shows best unlearnability on the poisoned class and keep decent performance on other classes. Therefore, by only perturbing one class of dataset (10% in CIFAR-10, 1% in CIFAR-100 and ImageNet-subset), UEs can make this class be unlearnable, without effecting model's ability for other classes.

Table 17 presents the experimental results of one-class UEs (only poisoned class 0) on CIFAR-100. Similar to CIFAR-10, our MI-UE still obtains the best unlearnabilty for targeted Class 0, while

Table 18: The unlearnable class accuracy and other classes accuracy of our MI-UE on ImageNet-subset.

| Test Acc(%) | Unlearnable Class | Other Classes |
|---|---|---|
| Clean | 92 | 80.31 |
| MI-UE | 16 | 72.61 |

keeping the accuracy on other classes. It is worth noting that some UEs, namely AR and GUE, become ineffective when only poisoned one class on CIFAR-100. This may because CIFAR-100 has 100 classes, poisoned one class for CIFAR-100 only give room for 1% poisoned ratio, making the UEs more difficult. Therefore, some unstable UEs will fail to achieve their unlearnability.

Table 18 shows the performance of our MI-UE under one-class perturbations on ImageNet-subset. It displays similar trends from those on CIFAR-10/100, our MI-UE can destroy the generalization of the poisoned class while maintain decent accuracy of other classes.

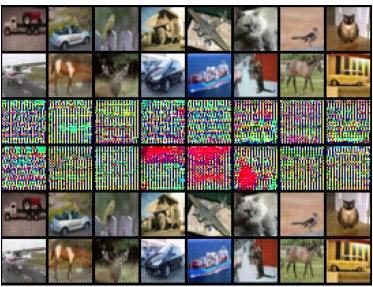

Figure 4: Visualization of MI-UE unlearnable noises and their corresponding clean and poisoned images on CIFAR-10. The first row is the clean images, the second row is the MI-UE noises, the last row is the poisoned images.

### G.2 VISUALIZATION

We display some clean images, MI-UE unlearnable noises and corresponding poisoned images of CIFAR-10 and ImageNet-subset dataset in Figures 4 and 5. The MI-UE noises are normalized to $[0, 1]$ for visualization. It can be seen that MI-UE noises are relatively regular than random noises, showing certain local isotropy.

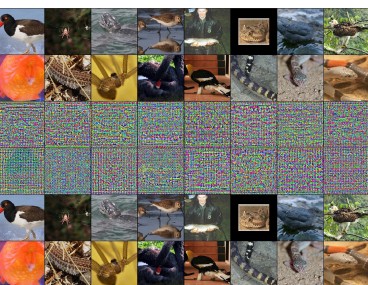

Figure 5: Visualization of MI-UE unlearnable noises and their corresponding clean and poisoned images on ImageNet-subset. The first row is the clean images, the second row is the MI-UE noises, the last row is the poisoned images.

### G.3 MORE RESULTS ON TRANSFERABILITY

The test accuracy of various baseline UEs, including EM, AP, NTGA, AR, REM, SEM, GUE and TUE, corresponding with our proposed MI-UE on CIFAR-100 dataset are provided in Table 19. The

results show that MI-UE obtains superiority on ResNet-18, ResNet-50, DenseNet-121, WRN34-10 and ViT-B. Therefore, MI-UE demonstrates the superior transferability across six modern deep networks, leading to the corresponding UE most successful.

Table 19: Test accuracy(%) of various UEs under different victim models on CIFAR-100. Our MI-UE achieves the lowest test accuracy compared to other UEs, indicating excellent poisoning effectiveness.

| Model/Method | Clean | EM | AP | NTGA | AR | REM | SEM | GUE | TUE | MI-UE (ours) |
|---|---|---|---|---|---|---|---|---|---|---|
| ResNet-18 | 76.65 | 2.09 | 3.73 | 3.08 | 6.19 | 7.52 | 6.29 | 22.79 | 1.34 | **1.17** |
| ResNet-50 | 78.25 | 2.14 | 4.51 | 5.21 | 9.05 | 7.63 | 4.55 | 23.51 | 3.91 | **1.72** |
| DenseNet-121 | 77.78 | 2.69 | 3.90 | 6.04 | 5.26 | 7.63 | 4.54 | 24.35 | 2.10 | **1.11** |
| WRN34-10 | 80.46 | 2.45 | 3.16 | 3.37 | 5.21 | 5.77 | 5.00 | 31.21 | 4.64 | **1.48** |
| ViT-B | 66.54 | 4.25 | 3.23 | 11.52 | 24.10 | 8.44 | 8.80 | 24.01 | 9.42 | **2.62** |

### G.4 MI-UE UNDER CONTRASTIVE LEARNING

Although our MI-UE is designed on supervised learning paradigm, we can also extend it to constrative learning paradigm directly. Inspired by (Wang et al., 2024b), we incorporate stronger contrastive augmentation into our MI-UE, denoted as A-MI-UE. Results in Table 20 show that the modified A-MI-UE outperforms both EM and contrastive UEs, TUE (Ren et al., 2022), on SimCLR (Chen et al., 2020), achieving stronger contrastive unlearnability.

Table 20: The performance of our A-MI-UE method compared with UE and TUE on SimCLR contrastive learning paradigm.

| SimCLR | Clean | EM | TUE | A-MI-UE |
|---|---|---|---|---|
| Test Acc(%) | 91.71 | 89.54 | 56.32 | **52.84** |

### G.5 LINEAR SEPARABILITY OF VARIOUS UNLEARNABLE NOISES AND DATASETS

We evaluate the linear separability of both unlearnable noises and unlearnable datasets for various existing methods. Specifically, we check the training accuracy of noise dataset $\{\epsilon_i, y_i\}_{i=1}^N$ fitted by the linear network as the result of Unlearnable Noises, and check the training accuracy of unlearnable dataset $\{x_i + \epsilon_i, y_i\}_{i=1}^N$ fitted by the linear network as the result of Unlearnable Datasets. It is noteworthy that we do not include any data augmentations when training the linear network, like Random Crop and Random Horizontal Flip, as suggested in Zhu et al. (2024b). The results are shown in Table 21. It reveals that although many unlearnable noises and datasets indeed have decent linear separability, some effective unlearnable methods like AP and AR demonstrate poor linear separability, especially for the AR method, their linear separability is almost approaching to the clean dataset. To quantify the correlation between linear separability and unlearnable power, we evaluate the Spearman correlation score between Acc gap with Training Acc on both Unlearnable Noises and Unlearnable Datasets, the score is 0.0333 and 0.2833 respectively, significantly lower than the score between Acc gap and MI gap, 0.7818.

### G.6 VARIOUS TRAINING STEP SIZES

To further demonstrate the soundness of our proposed MI-UE, we conduct experiments on CIFAR-10/100 with various poison generation step sizes, including 0.1/255, 0.2/255 (default), 0.4/255 and 0.8/255. Results in Table 22 show that different step sizes do not affect the unlearnable power of MI-UE.

Table 21: The training accuracy of unlearnable noises and unlearnable datasets by the linear network.

| Training Acc(%) | Unlearnable Noises | Unlearnable Datasets |
|---|---|---|
| Clean | — | 47.99 |
| EM | 99.32 | 99.37 |
| AP | 86.53 | 56.96 |
| NTGA | 99.94 | 95.02 |
| AR | 42.09 | 48.13 |
| SEM | 96.66 | 83.01 |
| REM | 92.97 | 81.46 |
| TUE | 100.0 | 100.0 |
| GUE | 98.92 | 99.63 |
| MI-UE (ours) | 99.89 | 99.91 |

Table 22: Quantitative results(%) of MI-UE with different poison generation step sizes.

| Step size | CIFAR-10 | CIFAR-100 |
|---|---|---|
| 0.1/255 | 9.93 | 1.13 |
| 0.2/255 (default) | 9.95 | 1.17 |
| 0.4/255 | 10.00 | 1.14 |
| 0.8/255 | 9.96 | 1.09 |

## G.7 COMPUTATIONAL COST

On CIFAR-10 and CIFAR-100, the generation of MI-UE requires about 3.6 hours. On ImageNet-subset, the generation of MI-UE requires about 45 hours. All of the experiments are conducted on a single NVIDIA A800 GPU.

Due to incorporation of the similarity matrix of our MI reduction loss $\mathcal{L}_{mi}$, the generation of poisons results in computational overhead to a certain extent, especially for larger-resolution datasets like ImageNet.

To mitigate the potential computational overheads, we test MI-UE with smaller poisoning epochs, 15 epochs and 30 epochs, for ImageNet-subset. Results are provided in Table 23. Compared with Table 2, MI-UE under 30 poisoning epochs demonstrates the state-of-the-art unlearnability across existing UEs. Even for 15 poisoning epochs, MI-UE still achieves the second-best performance, slightly behind EM. The effectiveness of MI-UE under economic scenarios further demonstrate MI-UE's real-world applications.

Table 23: Test accuracy of MI-UE with different poisoning epochs for ImageNet-subset.

| Epochs | Test Acc(%) |
|---|---|
| 50 (baseline) | 1.03 |
| 30 | 1.09 |
| 15 | 1.95 |

## G.8 LEARNING PROCESS

In this section, we visualize the evolution of training and test accuracies of our MI-UE, two representative baseline UEs, EM and AP, and two robust UEs, REM and SEM, on CIFAR-10 dataset. Figure 6 shows the learning process at each epoch for standard training. It suggests that the unlearnability of EM and REM are relatively weak, the training accuracy for SEM is relatively unstable. AP and our MI-UE hold both stable learning process and decent performance for standard training, and our MI-UE is slightly outperforming AP while displaying more stable test accuracy than AP at different epochs.

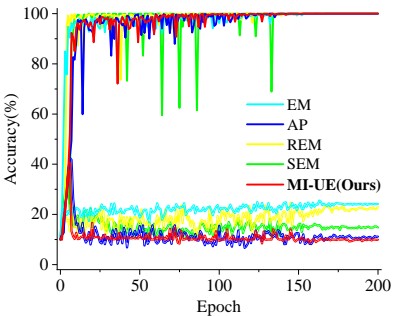

Figure 6: Learning process of standard training on EM, AP, REM, SEM and our MI-UE unlearnable examples, the solid lines represent the training accuracy, while the hollow lines represent the test accuracy.

Meanwhile, Figures 7, 8, 9, and 10 represents the learning process for adversarial training with defense budget be $8/255$, $6/255$, $4/255$ and $2/255$. Figures 7 and 8 reveals the learning process under larger adversarial training budget, suggesting the superiority of our MI-UE. The proposed robust UEs, REM and SEM, lose their unlearnability when the adversarial budget is larger than $1/2$ of poisoned budget. Figure 9 represents the learning process under adversarial training budget be $1/2$, in that case robust UEs (REM and SEM) work well, but traditional UEs (EM and AP) still be poor. Our MI-UE achieves comparable unlearnability with existing state-of-the-art UE, namely SEM, showing strong stability of our method. Figure 10 represents learning process under smaller adversarial training budget, in that case not only robust UEs, but also traditional UEs have shown unlearnability. Our MI-UE keeps the unlearnabilty in this scenario, achieve the best performance compared with existing methods.

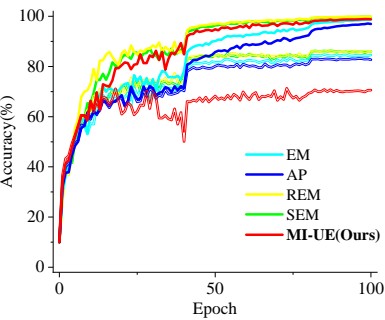

Figure 7: Learning process of adversarial training on EM, AP, REM, SEM and our MI-UE unlearnable examples with the perturbation budget be $8/255$, the solid lines represent the training accuracy, while the hollow lines represent the test accuracy.

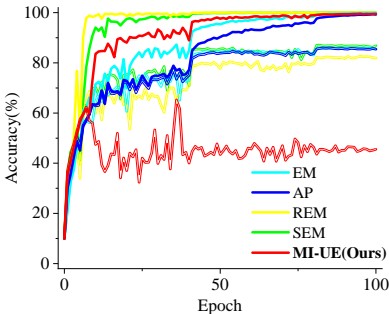

Figure 8: Learning process of adversarial training on EM, AP, REM, SEM and our MI-UE unlearnable examples with the perturbation budget be $6/255$, the solid lines represent the training accuracy, while the hollow lines represent the test accuracy.

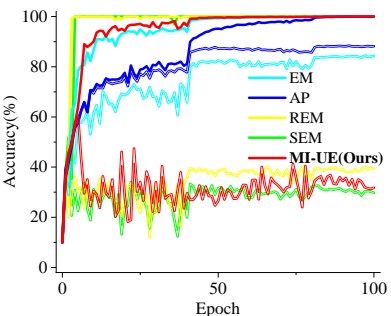

Figure 9: Learning process of adversarial training on EM, AP, REM, SEM and our MI-UE unlearnable examples with the perturbation budget be $4/255$, the solid lines represent the training accuracy, while the hollow lines represent the test accuracy.

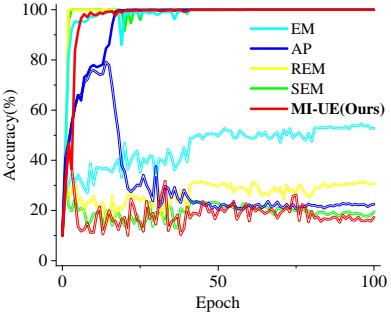

Figure 10: Learning process of adversarial training on EM, AP, REM, SEM and our MI-UE unlearnable examples with the perturbation budget be $2/255$, the solid lines represent the training accuracy, while the hollow lines represent the test accuracy.

