# OpenReview forum: "Why Do Unlearnable Examples Work: A Novel Perspective of Mutual Information"
_ICLR.cc/2026/Conference — ICLR 2026 Poster_

### Official Review · Reviewer_L5MP · 2025-10-17

**Soundness:** 2
**Presentation:** 4
**Contribution:** 3
**Rating:** 6
**Confidence:** 5

**Summary:**

The paper argues that the effectiveness of unlearnable examples is heuristic and not well explained, and they set out to show that the mutual information gap presented by unlearnable datasets can better explain the effectiveness of reducing test accuracy. The authors propose a new optimization objective which reduces the covariance of intraclass features given the labels. Extensive evaluations show this approach is better than prior approaches in terms of its reduction of test accuracy, effectiveness against defenses, and transferability.

**Strengths:**

1. MI-UE is evaluated on 3 classic datasets and compared with classic unlearnable examples, demonstrating that it works best across datasets (Table 2), transfers best from RN-18 to other models (Table 3), works the best across adversarial training (Table 4), and across other defense methods (Table 5). Overall, I am impressed with how consistently this approach works.
2. The fact that these unlearnable examples work so well lend credence to their hypothesis that mutual information plays a key role (though, this can be improved and is discussed in weaknesses section)
3. Good ablations show that the similarity term in their loss is more important than the distance term (Table 6).

**Weaknesses:**

1. Motivation: The author's argument around linear separability is that "we find that linear classifiers trained on UEs can achieve certain generalization...interpreting UEs merely as linear shortcuts does not fully account for why such examples result in worse generalization". But prior studies like [1] study the linear separability of **perturbations**, not of the poisoned images. The language implies these linear models were not trained on just perturbations.
2. Using Table 1, the authors compute a correlation between the MI gap and Acc Gap of 0.78. This is good but in order to really show that "effective unlearnable examples **always decrease** mutual information" other alternatives should also be considered. For example: one could consider the linear separability of the perturbation (e.g. train a linear layer on perturbations) and correlate that to the Acc Gap. Is that correlation higher or lower? This would be a start (and necessary since the authors talk about linear separability in motivation), but it would be nice to have another metric if possible too. The authors could use prior work to find other "explanations" for why UEs work.
3. L251 "the reduction of MI is the primary cause". I think I agree that MI could be a cause, but to show it is the primary cause the authors will need to consider alternatives (See previous point above).
4. Regarding ISS [3], more details need to be given as to how the evaluation was performed. ISS paper proposed JPEG and grayscale as separate augmentations that broke a lot of unlearnable methods. I am particularly interested in an evaluation of *just ISS* (different JPEG compression qualities must be tried: 0.9, 0.8, 0.7, etc) instead of all the other "defenses" (cutout, cutmix, mixup, OP, etc.) because JPEG was shown to be so effective.

Typos:
- "correlation score be 0.7818" -> "of 0.7818"

[1] Availability attacks create shortcuts. Yu et al., 2022

[3] Image shortcut squeezing: Countering perturbative. Liu et al. 2023

**Questions:**

1. Above all, my suggestion is to work on the weaknesses described above.
2. I don't understand the message/point of Figure 1. For example, what is the histogram-based estimator estimating? The paragraph starting at L206 says that Fig. 1 is "To further ensure the confidence of MI estimation" but how? Should be described at least at a high level - what does this figure add?
3. The MI-UE noise in G.2. looks interesting. I wonder if these are linearly separable: if one takes the perturbations only (for each class) and trains a linear classifier on the perturbations, what accuracy is achieved?
4. I'm curious what the train loss / test accuracy curves for MI-UE look like when training on CIFAR-10. In particular, I wonder if there are points during training when test accuracy peaks like some poisons do in [2]

[2] What can we learn from unlearnable datasets?, Sandoval-Segura et al., 2023

---

> ### Author Response · Authors · 2025-11-21
> **Rebuttal**
>
> Thanks for your positive feedback and insightful comments. Below we address the detailed comments, and hope that you can find our response satisfactory.
>
> **Question 1: Prior studies like [1] study the linear separability of perturbations, not of the poisoned images…**
>
> Indeed, [1] only studies the linear separability of unlearnable noises. However, [a] has expanded this linear separability from noises to datasets. Specifically, Theorems 1 and 2 in [a] have proved that certain unlearnable datasets are linearly separable. Beyond that, they propose to use linear networks to detect the existence of unlearnable examples. We have made our statement clearer in the updated version (in blue). Furthermore, to empirically demonstrate this fact, we also add additional experiments on the linear separability of both unlearnable noises and datasets in our Answer 2 and Appendix G.5 in our updated manuscript.
>
> Ref:
>
> [a] Zhu et al. Detection and defense of unlearnable examples. AAAI 2024.
>
> **Question 2: One could consider the linear separability of the perturbation (e.g. train a linear layer on perturbations) and correlate that to the Acc Gap. Is that correlation higher or lower…**
>
> Thanks for your valuable advice. We evaluate the linear separability of both unlearnable noises and unlearnable datasets for various existing methods, the results are shown below.
>
> |Training Acc(\%)| Unlearnable Noises | Unlearnable Datasets|
> |-|-|-|
> |Clean | --- | 47.99 |
> |EM  | 99.32 | 99.37 |
> |AP |  86.53 | 56.96 |
> |NTGA | 99.94 | 95.02 |
> |AR  | 42.09 | 48.13 |
> |SEM | 96.66 | 83.01 |
> |REM | 92.97 | 81.46 |
> |TUE | 100.0 | 100.0 |
> |GUE |  98.92 | 99.63  |
> |MI-UE (ours) | 99.89 | 99.91|
>
> It reveals that although many unlearnable noises and datasets have decent linear separability, some effective unlearnable methods like AP and AR have poor linear separability, especially for the AR method, their linear separability is almost approaching to the clean dataset. To quantify the correlation between linear separability and unlearnable power, we evaluate the Spearman correlation score between Acc gap with Training Acc on both Unlearnable Noises and Unlearnable Datasets, the score is 0.0333 and 0.2833 respectively, significantly lower than the score between Acc gap and MI gap, 0.7818. Therefore, we believe that MI reduction is the primary cause of unlearnability rather than the linear separability of noises/datasets. We have added this evaluation into our Appendix G.5 in the revised version.
>
>
> **Question 3: I am particularly interested in an evaluation of just ISS (different JPEG compression qualities must be tried: 0.9, 0.8, 0.7, etc) instead of all the other "defenses" (cutout, cutmix, mixup, OP, etc.) because JPEG was shown to be so effective.**
>
> Thanks for your suggestion. The standard JPEG compression of ISS is 10. We select the JPEG compression in ISS with different strengths 6, 8, 10, 12, 15, 20 and 30 as the defense of unlearnable examples, the quantitative results are provided in the following table.
>
> |Quality | 6 | 8 | 10 | 12 | 15 | 20 | 30 |
> |-|-|-|-|-|-|-|-|
> |Clean | 78.52 | 81.35 | 83.12 | 83.97 | 85.41 | 86.57 | 88.17 |
> |EM | 77.57 | 80.96 | 82.94 | 83.63 | 84.91 | 85.67 | 86.30 |
> |NTGA | 76.39 | 78.63 | 79.38 | 79.64 | 81.68 | 81.59 | 81.50 |
> |AR | 78.89 | 81.55 | 82.83 | 84.18 | 85.39 | 86.68 | 88.41 |
> |REM | 78.04 | 81.30 | 82.20 | 83.23 | 84.50 | 84.60 | 85.24 |
> |SEM | 77.91 | 80.36 | 82.14 | 81.79 | 81.16 | 78.01 | 69.36 |
> |GUE | 78.27 | 80.69 | 83.00 | 83.97 | 85.39 | 86.46 | 88.06 |
> |TUE | 78.48 | 81.83 | 83.23 | 84.13 | 85.12 | 85.88 | 86.38 |
> |MIUE | 77.55 | 81.15 | 82.52 | 83.92 | 85.38 | 86.44 | 84.99 |
>
> NTGA outperforms on these JPEG quality compressions. Our MI-UE are comparable with REM, achieving the second-best unlearnable performance on average of these JPEG compressions quality. All these unlearnable examples become ineffective against JPEG compression, with accuracy recovery to over 80\%, indicating that bypassing tailored defense for UEs remains a challenged problem. We have complemented these evaluations in Section 6.4 of our updated paper.

---

> ### Author Response · Authors · 2025-11-21
> **Rebuttal**
>
> **Question 4: I don't understand the message/point of Figure 1. For example, what is the histogram-based estimator estimating? The paragraph starting at L206 says that Fig. 1 is "To further ensure the confidence of MI estimation" but how? Should be described at least at a high level - what does this figure add?**
>
> In Figure 1, these MI estimators evaluate the MI between clean and unlearnable poisoned features, i.e., $I(g(X), g(X’))$.
>
> Only use one type of MI estimator may be not convincing as estimating MI is not easy for high-dimensional data. Therefore, to further enhance the soundness of MI estimation, we deploy multiple MI estimators. Figure 1 shows the similar trends of these MI estimators, typically, every effective UE show MI reduction under all of these estimators. Therefore, Figure 1 further increases the confidence of our claim, UEs are caused by MI reduction.
>
> We have added these explanations in Figure 1 and Section 4 of our revised manuscript.
>
> **Question 5: The linear separability of MI-UE noises…**
>
> We have evaluated the linear separability of MI-UE noises and datasets in our Answer 2 and updated Appendix G.5. In detail, the training accuracy of MI-UE noises under the linear network is 99.89\%, showcasing strong linear separability. However, as we discussed in Answer 2, although many of existing UEs, including our MI-UE, demonstrating linear separability, the Spearman correlation score between them and Acc Gap is still very low (0.0333 for UE noises linear separability). Therefore, we believe it is MI reduction rather than linear separability induces unlearnability.
>
> **Question 6: What the train loss / test accuracy curves for MI-UE look like when training on CIFAR-10…**
>
> We have provided the training curve of various UEs, including our MI-UE in Figure 6 of Appendix G.7. Specifically, the peak accuracy of MI-UE is 40.56\%, which occurs in the 6th epoch.
>
> **Question 7: "correlation score be 0.7818" -> "of 0.7818".**
>
> Thanks for pointing out the typo in our manuscript. We have corrected it in the updated version.

---

### Official Review · Reviewer_nUBW · 2025-10-29

**Soundness:** 2
**Presentation:** 2
**Contribution:** 2
**Rating:** 2
**Confidence:** 4

**Summary:**

This paper studies how to generate unlearnable examples better. The authors propose to reduce the mutual information between the original examples and the generated data. The paper claims that the method is supported by a theorem. Extensive experimental results are presented.

My first concern is the novelty and significance of reducing the mutual information between original examples and generated data to generate unlearnable examples. The same idea has been in [1]. A large volume of literature is on similar methods based on KL divergence.

Second, the theory relies on an unjustified assumption that the poison distribution is "is close to a Gaussian mixture distribution," which makes the theory unconvincing.

In the experiments, many settings are not justified; e.g., "we set the total poison budget at 8/255," "a step size of 0.2/255," "the step size of PGD be 0.4/255," etc. I would suggest that the authors give results on the hyperparameter sensitivity.

I didn't find how many trials the test run. If the reported results are from a single run, I would like to question their credibility. From my experiences, the results likely have considerable randomness. I would suggest the authors report (a) averages and (b) standard deviations of multiple (ideally over 5) trials.

The presentation can be improved.

[1] Wang, B., Tian, J., Wang, X., Yuan, X. and Li, J., 2025. Reversible Unlearnable Examples: Towards the Copyright Protection in Deep Learning Era. IEEE Transactions on Circuits and Systems for Video Technology.

**Strengths:**

Please see above.

**Weaknesses:**

Please see above.

**Questions:**

Please see above.

---

> ### Author Response · Authors · 2025-11-21
> **Rebuttal**
>
> Thanks for your valuable and insightful comments. Below we address the detailed comments, and hope that you can find our response satisfactory.
>
> **Question 1: The novelty and significance of reducing the mutual information between original examples and generated data to generate unlearnable examples. The same idea has been in [1].**
>
> We kindly remind that the reference paper [1] is published on 08 October 2025, later than the submission deadline of ICLR 2026, 24 September 2025.
>
> Furthermore, although [1] also tries to generate unlearnable examples by minimizing the mutual information, the core idea, motivation and methodology are totally different. First, from the perspective of the core idea, [1] aims to impede the learning process of victim model by minimizing MI between **model inputs** and **model outputs**, where our MI-UE is achieved by minimizing MI between **clean features** and **poison features**, the two are completely different. Inspired by information theory, we propose a novel and more theoretically grounded framework for generating unlearnable examples. Specifically, we observe the strong correlation between MI reduction and accuracy drop, and further provide a theorem in Appendix A, showcasing that the generalization upper bound increases as the MI between clean and poisoned features decrease. In contrast, [1] just optimizes MI of model inputs and outputs heuristically, without theoretical analyses or concrete correlation measurement.
>
> Second, from the perspective of methodology, [1] optimizes the MI between model inputs and outputs by minimizing the entropy of model outputs, where our MI-UE optimizes the MI between clean and poisoned features by maximizing the similarity for intra-class features under the theoretical guarantee provided in Theorem 5.1. Therefore, compared with [1], our MI-UE has novelty on the core idea, motivation and methodology.
>
> Moreover, our MI-UE achieves better unlearnability than [1] in both CIFAR-10 (9.95\% v.s. 10.50\%) and ImageNet-100 (1.03\% v.s. 2.03\%) as shown in our Table 1 and Table II in [1], making our MI-UE more significant. We have updated our manuscript to include [1] in the related work in Section 2.
>
> In addition, although some work like [a] has utilized KL-divergence to construct UE, it tries to prevent knowledge distillation from authorized network by increasing the KL-divergence between features of authorized and hacker (unlearnable) networks. Differently, our MI-UE doesn’t need two networks, we achieve the state-of-the-art unlearnability by minimizing MI of clean and poisoned features of the same unlearnable network. The two are totally different.
>
> Overall, we are the first paper to utilize MI between clean and poisoned features to explain why UEs work, and provide an effective algorithm to achieve the state-of-the-art unlearnability. Therefore, we believe our MI-UE has both novelty and significance.
>
>
> Ref:
>
> [1] Wang et al. Reversible Unlearnable Examples: Towards the Copyright Protection in Deep Learning Era. **Published on 08 October 2025 in IEEE TCSVT.**
>
> [a] Ye and Wang. Ungeneralizable Examples. CVPR 2024.
>
>
> **Question 2: The theory relies on an unjustified assumption that the poison distribution is "is close to a Gaussian mixture distribution"…**
>
> It is reasonable to assume some Gaussian distribution when conducting theoretical analyses for data poisoning, like [b,c,d,e]. From an empirical perspective, modern neural networks in feature space, after normalization operations such as BatchNorm/LayerNorm, often present a nearly isotropic distribution, which has a similar covariance structure to the Gaussian distribution [f].
>
> Even if the actual distribution deviates from Gaussian (with $\epsilon$ KL divergence gap), the upper bound given by Theorem 5.1 is still a function of $\Sigma_Y$, and our MI-UE actually optimizes by maximizing the cosine similarity of similar features to compress the intra-class covariance. Therefore, MI-UE doesn’t rely on strict Gaussianity, as long as the inter-class/intra-class covariance is controllable, it can play a role in MI reduction and achieving unlearnability.
>
> Ref:
>
> [b] Sadasivan et al. CUDA: Convolution-based Unlearnable Datasets. CVPR 2023.
>
> [c] Wang et al. Robust Learning for Data Poisoning Attacks. ICML 2021.
>
> [d] Wang et al. Lethal Dose Conjecture on Data Poisoning. NeurIPS 2022.
>
> [e] Suya et al. What Distributions are Robust to Indiscriminate Poisoning Attacks for Linear Learners? NeurIPS 2023.
>
> [f] Daneshmand et al. Batch Normalization Orthogonalizes Representations in Deep Random Networks. NeurIPS 2021.

---

> ### Author Response · Authors · 2025-11-21
> **Rebuttal**
>
> **Question 3: In the experiments, many settings are not justified…I would suggest that the authors give results on the hyperparameter sensitivity.**
>
> Thanks for your advice. We set the poisoning budget $\epsilon=8/255$ as default to make sure a fair comparison, following all existing UE methods under $L_{\infty}$ norm. To validate our MI-UE in broader scenarios, we further generate MI-UE with different poisoning budget from $4/255$ to $16/255$, and evaluate the unlearnability in the following table. Results demonstrate that even though poisoning budgets across from $4/255$ to $16/255$, MI-UE always results in the unlearnability to a random guess level (i.e., 10% for CIFAR-10, 1% for CIFAR-100).
>
> |Budgets/Test Acc(\%)  | CIFAR-10 | CIFAR-100|
> |-|-|-|
> |4/255 | 10.49 | 1.16|
> |6/255 | 10.09 | 1.19 |
> |8/255 (default)  | 9.95 | 1.17 |
> |12/255 | 9.83 | 1.17|
> |16/255 | 9.97 | 1.09|
>
> For the step size in the training of unlearnable poisons, we conduct experiments on CIFAR-10/100 with various poison generation step sizes, including 0.1/255, 0.2/255 (default), 0.4/255 and 0.8/255. Results in the following table show that different step sizes do not affect the unlearnable power of MI-UE.
>
> |Step size   | CIFAR-10 | CIFAR-100 |
> |-|-|-|
> |0.1/255 | 9.93 | 1.13 |
> |0.2/255 (default) | 9.95 | 1.17 |
> |0.4/255 | 10.00 | 1.14 |
> |0.8/255 | 9.96 | 1.09 |
>
> Furthermore, we have studied the balancing hyperparameter $\zeta$ in MI-UE loss, the result has been provided in Table 7. It demonstrates that $\zeta$ be 0.1 or less will slightly increase the effectiveness of MI-UE. Moreover, we also test MI-UE with different poisoning epochs for CIFAR-10 and CIFAR-100 in the following table.
>
> |Epochs/Test Acc(\%)    | CIFAR-10 | CIFAR-100|
> |-|-|-|
> |100 (default)  | 9.95 | 1.17 |
> |50  | 10.25 | 1.66 |
> |20  | 15.39 | 3.23 |
>
> Results demonstrate that with half of the generation time (50 epochs), MI-UE still outperforms on CIFAR-10 and achieves the second-best performance on CIFAR-100 compared with existing UEs shown on Table 2 in Section 6. Furthermore, MI-UE still gains comparable results with other UEs even the poisoning epochs are reduced to 20.
>
> We have updated these results (in blue) about ablation studies for hyperparameters on Section 6.4 in the new version of our paper.
>
>
>
> **Question 4: I would suggest the authors report (a) averages and (b) standard deviations of multiple (ideally over 5) trials.**
>
> Thanks for your suggestion. To make sure the evaluation across different methods are stable and persuasive, we run each unlearnable examples on CIFAR-10 and CIFAR-100 for 5 times on different random seeds, and report the mean test accuracy as well as the stand error. The evaluation results are provided in the following:
>
> | Method/Test Acc(\%) | CIFAR-10 | CIFAR-100 |
> |-|-|-|
> |Clean|94.51$\pm$0.19| 1.15$\pm$0.06 |
> |EM|23.88$\pm$1.89| 2.03$\pm$0.43 |
> |AP|11.29$\pm$0.83| 3.85$\pm$1.47
> |NTGA|23.06$\pm$1.18| 3.04$\pm$0.55 |
> |REM|22.88$\pm$0.68| 7.37$\pm$1.10|
> |SEM|14.81$\pm$0.65| 4.21$\pm$0.67|
> |TUE|11.28$\pm$0.31| 4.94$\pm$0.37 |
> |MI-UE(ours)|**9.95$\pm$0.04**| **1.05$\pm$0.05**|
>
> It shows that our MI-UE indeed outperforms other baseline methods under multiple runs on different random seeds, further demonstrating the effectiveness of our proposed method.
>
>
> **Question 5: The presentation can be improved.**
>
> Thanks for your advice. We will further improve our presentations in the revised version of our manuscript.

---

### Official Review · Reviewer_i275 · 2025-11-01

**Soundness:** 3
**Presentation:** 3
**Contribution:** 3
**Rating:** 6
**Confidence:** 4

**Summary:**

This paper introduces a novel approach for generating unlearnable examples, focusing on mutual information (MI) reduction as the mechanism behind their effectiveness. The authors analyze the relationship between MI reduction and the unlearnability of examples, providing both theoretical and experimental results. They propose a new method called Mutual Information Unlearnable Examples (MI-UE), which aims to reduce MI by optimizing the covariance between poisoned and clean features. The paper presents experiments comparing MI-UE to state-of-the-art methods and demonstrates its superior performance, particularly in reducing model generalization ability.

**Strengths:**

- The paper presents a interesting perspective on unlearnable examples, introducing MI reduction as the key mechanism. This approach is novel and provides a theoretical understanding of how UEs work, going beyond empirical heuristics.

- The paper includes theoretical grounding, demonstrating a connection between MI reduction and unlearnability. The experiments are well-designed, comparing MI-UE with several baseline methods and across different model architectures.

- The paper is well-written, with clear definitions, logical flow, and accessible explanations. Complex concepts like MI and covariance reduction are explained systematically.

**Weaknesses:**

- The paper acknowledges the challenges in optimizing MI and covariance, but the proposed solution still relies on a relatively simple optimization process. Further exploration into alternative optimization strategies could strengthen the paper's impact.

- The bi-level optimization lacks sufficient theoretical analysis. Given that first-order gradient-based methods for solving bi-level optimization problems face well-known convergence challenges in non-convex scenarios, the absence of theoretical or empirical investigation diminishes the soundness of the method. It would be beneficial for the authors to substantiate the convergence of the proposed method through theoretical analysis or empirical validation.

- Lack of novelty in experimental setups. While the theoretical contribution is novel, the experiments are primarily based on standard benchmark datasets (CIFAR-10, CIFAR-100, and ImageNet-subset), which are commonly used in related work. This reduces the perceived novelty of the empirical evaluation.

- While the authors claim that MI-UE significantly outperforms existing methods, the computational complexity of the proposed method is not fully discussed. It would be useful to have a more detailed analysis of how MI-UE scales with large datasets and models.

**Questions:**

1. Could the authors elaborate on the scalability of MI-UE when applied to large datasets or deep models beyond the tested architectures? (e.g. LLMs)

2. How does MI-UE perform in terms of convergence? Does MI-UE deliver a superior computation–utility trade-off compared with existing methods?

---

> ### Author Response · Authors · 2025-11-21
> **Rebuttal**
>
> Thanks for acknowledging the novelty of our paper and providing positive feedback. Below we address the detailed comments, and hope that you can find our response satisfactory.
>
> **Question 1: The proposed solution still relies on a relatively simple optimization process. Further exploration into alternative optimization strategies could strengthen the paper's impact.**
>
> Thanks for your valuable suggestion. Our MI-UE raises from Theorem 5.1, which proves that minimizing MI can be implicitly achieved by minimizing the conditional covariance of intra-class features. Then we optimize the similarity of both intra-classes and inter-classes features to induce the effective algorithm. To make the optimization easy and clear, we iteratively optimize model $\theta$ and poison $\delta$ for the bi-level problem in Eq. (3) just as previous work (EM, REM, TUE, etc.). Noting that the poison $\delta$ is what we really want, the parameter $\theta$ is only used as the shadow model, a potential alternative optimization method is to optimize the shadow model $\theta$ several times (epochs), then optimize the poison $\delta$ for one time. This approach may increase the stability of generated poisons. But the potential drawback is that this process may require heavier computational cost for an effective poison $\delta$, as we need to optimize more iterations for the shadow model $\theta$.
>
> Moreover, beyond our MI reduction approach, we have also studied to directly introduce a MI regularizer into the loss function in Sec 6.4. As shown in Table 8, although MI regularizers can also improve the MI reduction and unlearnability, they still fall behind to our MI-UE approach. More alternative algorithms for MI reduction can be leaved as the future work.
>
> |Method|Acc(Acc Gap)(\%)|MI(MI Gap)|
> |-|-|-|
> |Clean(baseline)|94.45|0.7122|
> |UE|24.17(70.28)|0.6400(0.0722)|
> |UE+MI reg.|15.62(78.83)|0.5336(0.1786)|
> |AP|11.21(83.24)|0.5871(0.1251)|
> |AP+MI reg.|10.01(84.44)|0.5183(0.1939)|
> |MI-UE(Ours)|**9.95**(**84.50**)|**0.4969**(**0.2153**)|
>
> **Question 2: The bi-level optimization lacks sufficient theoretical analysis…**
>
> We acknowledge that the rigorous theoretical guarantee for bi-level optimization under non-convex scenarios remain an unsolved challenge. However, intuitively, our MI-UE approach may be easier to optimize than traditional EM approach for poison $\delta$. EM approach optimizes both $\theta$ and $\delta$ with the same cross-entropy loss function $L_{ce}$, and our MI-UE approach optimizes $\theta$ with $L_{ce}$, optimizes $\delta$ with $L_{mi}$. In EM approach, if one term is easy to be fitted, the other term may suffer from degeneration problem. For instance, when EM first optimizes $\theta$ with $L_{ce}$, if the optimizer quickly finds a good point with very low loss $L_{ce}$, the optimization of $\delta$ may suffer from degeneration problem resulting from gradient vanishing. MI-UE mitigates this problem as $\delta$ and $\theta$ are optimized with different loss function, $L_{mi}$ does not assign concrete label for each feature, inducing different optimization direction with $\theta$ optimization by $L_{ce}$.
>
> Furthermore, empirically, our MI-UE can obtain better unlearnable poison $\delta$ than existing UE methods. We have demonstrated the effectiveness of our MI-UE poison $\delta$ in Section 6, with the state-of-the-art performance on three benchmark datasets, various network architectures, adversarial training and many tailored defense methods. We believe the convergence and effectiveness of our bi-level MI-UE approach can be verified by these enormous experiments.

---

> ### Author Response · Authors · 2025-11-21
> **Rebuttal**
>
> **Question 3: Lack of novelty in experimental setups. The experiments are primarily based on standard benchmark datasets…**
>
> As existing UEs are usually evaluated on these benchmark datasets, we also test our MI-UE based on them to ensure a fair comparison. We want to emphasize that as an explanation work for why UEs work, make sure the fairness may be more important than introducing some novel datasets/scenarios.
>
> Nevertheless, to further enhance the novelty of our empirical evaluation, we extend our MI-UE to 3D point clouds. We conduct a preliminary experiment of MI-UE on ModelNet40 dataset [a] with PointNet [b] and DGCNN [c] as the victim model. Compared with baseline methods like UE and AP, MI-UE achieves the superior unlearnability under the scenario of 3D point clouds, reduces the test accuracy from over 90\% to about 30\%. We believe the efficacy of MI-UE under 3D point clouds can further demonstrate its applicability in more complex datasets and scenarios.
>
> | Method/ Test Acc(\%) | PointNet | DGCNN |
> |-|-|-|
> |Clean (baseline) | 90.81 | 92.59 |
> | UE | 82.35 | 80.51|
> | AP | 69.56 | 78.15 |
> |MI-UE| 28.35 | 32.01 |
>
>
> Ref:
>
> [a] Wu et al. 3D ShapeNets: A Deep Representation for Volumetric Shapes. CVPR 2015.
>
> [b] Qi et al. PointNet: Deep Learning on Point Sets for 3D Classification and Segmentation. CVPR 2017.
>
> [c] Wang et al. Dynamic Graph CNN for Learning on Point Clouds. ACM TOG 2019.
>
>
> **Question 4: The computational complexity of the proposed method is not fully discussed. Does MI-UE deliver a superior computation–utility trade-off compared with existing methods? It would be useful to have a more detailed analysis of how MI-UE scales with large datasets and models.…**
>
> We have provided the computational cost in Appendix G.6, all of our experiments are conducted on a single NVIDIA A800 GPU. Due to incorporation of the similarity matrix of our MI reduction loss, our MI-UE introduce about 1.5 times computational overhead compared with baseline method like EM. An easy and effective approach to mitigate these overheads is to reduce the poisoning epochs. To validate the effectiveness, we test MI-UE with smaller poisoning epochs (50 and 20) for CIFAR-10 and CIFAR-100 in the following table.
>
> |Epochs/Test Acc(\%)    | CIFAR-10 | CIFAR-100|
> |-|-|-|
> |100 (default)  | 9.95 | 1.17 |
> |50  | 10.25 | 1.66 |
> |20  | 15.39 | 3.23 |
>
> Results demonstrate that with half of the generation time (50 epochs), MI-UE still outperforms on CIFAR-10 and achieves the second-best performance on CIFAR-100 compared with existing UEs shown on Table 2 in Section 6. Furthermore, MI-UE still gains comparable results with other UEs even the poisoning epochs are reduced to 20.
>
> Furthermore, for ImageNet-subset, we test our MI-UE with 30 poisoning epochs and 15 poisoning epochs. Compared with Table 2 in Section 6, MI-UE under 30 poisoning epochs demonstrates the state-of-the-art unlearnability across existing UEs. Even for 15 poisoning epochs, MI-UE still achieves the second-best performance, slightly behind EM.
>
> |Epochs/Test Acc(\%)     | ImageNet-subset |
> |-|-|
> |50 (baseline)  | 1.03 |
> |30 | 1.09 |
> |15  | 1.95 |
>
> The effectiveness of MI-UE under economic scenarios further demonstrate MI-UE’s real-world applications. Therefore, MI-UE does indeed offer a superior computation-utility trade-off. We add these additional discussions on smaller poisoning epochs into our Section 6.4 and Appendix G.6 in our revised manuscript.
>
> Furthermore, as our MI-UE only requires MI reduction between features, we believe MI-UE can easily expand to large datasets and models. A primary result shown in Appendix F.4 has demonstrated that MI-UE gains consist unlearnability and MI reduction when the network becomes deeper (from ResNet-18 to ResNet-152, from ViT-Base to ViT-Huge).
>
> |Method||ResNet-18|ResNet-34|ResNet-50|ResNet-101|ResNet-152|
> |-|-|-|-|-|-|-|
> |MI-UE|Acc(Acc Gap)(\%)|9.95 (84.50)|9.97(84.66)|9.98 (85.18)|9.99 (85.44)|9.90 (85.65)|
> ||MI(MI Gap)| 0.4969 (0.2153)|0.4857 (0.2210)|0.4802 (0.2276)|0.4621 (0.2401)|0.4338 (0.2697)|
>
> |Model| |ViT-Base | ViT-Large | ViT-Huge |
> |-|-|-|-|-|
> |MI-UE|Acc(Acc Gap)(\%)|15.51(75.41)|14.43(76.92)|13.78(78.15)|
> ||MI(MI Gap)|0.5332(0.1953)|0.5296(0.2001)|0.5180(0.2094)|

---

### Official Review · Reviewer_uCnz · 2025-11-02

**Soundness:** 3
**Presentation:** 2
**Contribution:** 3
**Rating:** 6
**Confidence:** 3

**Summary:**

This paper investigates why unlearnable examples (UEs) succeed at preventing unauthorized model training and introduces a mutual-information (MI)–based perspective to explain and enhance UE effectiveness. The authors empirically show that effective UEs consistently reduce the mutual information between clean and poisoned representations, and that deeper models suffer greater generalization degradation when trained on such data.
Building on this insight, they propose MI-UE, a new poisoning method that explicitly minimizes intra-class feature covariance and maximizes cosine similarity among poisoned features. Across CIFAR-10/100 and ImageNet-subset, MI-UE outperforms prior UE approaches under standard and adversarial training, and demonstrates improved transferability to various architectures.

**Strengths:**

1. Novel theoretical lens: Introduces mutual-information reduction as a unified explanation for UE effectiveness, filling a conceptual gap in a field often driven by heuristics.

2. Strong empirical evidence: Comprehensive experiments across datasets, models (CNNs and ViTs), and defense scenarios support claims.

3. Clear practical contributions: Proposed MI-UE method is simple, effective, and broadly transferable across architectures.

4. Good analysis of model depth effect: Demonstrates that deeper models are more vulnerable to UEs, which aligns with empirical observations and provides useful insight.

5. Solid ablations: Shows benefit of MI-driven optimization beyond simple feature clustering.

**Weaknesses:**

1. Limited analysis of failure modes and defenses.
While MI-UE achieves strong performance under many settings, it still degrades against certain tailored UE defenses (e.g., ISS, AVA) and does not universally dominate all specialized defense configurations. The paper acknowledges this but does not deeply examine why MI-UE fails in these cases, what properties of MI-based poisoning are being countered, or whether failure arises from optimization constraints, feature collapse dynamics, or assumptions about representation geometry. A more systematic characterization of failures and conditions under which MI-UE breaks down would strengthen the contribution.

2. High computational overhead and unclear scalability.
The proposed approach introduces bi-level optimization with inner-loop shadow model updates and iterative PGD steps, which may impose substantially higher compute cost than classic error-minimization-based UEs.
Although effective, this cost may limit usability in large-scale settings (e.g., ImageNet or web-scale self-supervised pipelines) where training even a single model is expensive. The paper does not report runtime cost, memory overhead, or scalability curves relative to baselines. It would be better to assess such practicality for real-world privacy preservation at large data scale.

**Questions:**

Please respond to Weaknesses.

---

> ### Author Response · Authors · 2025-11-21
> **Rebuttal**
>
> Thanks for acknowledging the novelty of our paper and providing positive feedback. Below we address the detailed comments, and hope that you can find our response satisfactory.
>
> **Question 1: Limited analysis of failure modes and defenses…A more systematic characterization of failures and conditions under which MI-UE breaks down would strengthen the contribution.**
>
> We believe these failures against SOTA defense are challenging problems against the whole area of UEs rather than just our MI-UE. For instance, [a] has already proved that UEs can be defended by adversarial training, if adversarial training budget is larger than UE poisoning budget. This is reasonable as if adversarial training budget is large enough, it will include the original clean data points even UEs exist. Therefore, a well adversarially trained network will learn the clean features (rather than only poisoned features), gain decent ability when they face with data from the clean distribution.
>
> Nevertheless, our MI-UE still achieves the best unlearnability in worst-case scenario (86.18\% under AVA), the second-best, SEM is 88.55\% under D-VAE, other UEs are over 90\% in worst-case defenses.
>
> For some effective tailored defense like ISS and AVA, to study the reason behind failure, we evaluate the quantitative results of MI and accuracy under ISS and AVA defense. The results provided below demonstrate that, although MI-UE still showcases larger MI gap compared with existing UEs, their gaps become significantly smaller, 0.0732 for ISS and 0.0948 for AVA. Recall the result in Table 1, MI-UE achieves 0.2153 MI gap in non-defense case. Therefore, stronger tailored defense like ISS and AVA, may implicitly weaken the MI reduction, inducing a smaller MI gap. This is a potential reason of the failure of MI-UE. Why tailored defenses can destroy MI reduction is an interesting topic, we leave it as the future work.
>
> |ISS|Clean|Random|EM|AP|NTGA|AR|REM|SEM|GUE|TUE|MI-UE|
> |-|-|-|-|-|-|-|-|-|-|-|-
> |Test Acc(\%)|82.71|82.66|82.78|82.50|80.84|82.79|82.59|81.86|83.10|82.61|81.35|
> |Acc Gap(\%)|-|0.05|-0.07|0.21|1.87|-0.08|0.12|0.85|-0.39|0.10|1.36|
> |MI|0.7059|0.6709|0.6680|0.6543|0.6552|0.6598|0.6722|0.6532|0.6688|0.6683|0.6327|
> |MI Gap|-|0.0350|0.0379|0.0516|0.0507|0.0461|0.0337|0.0527|0.0371|0.0376|0.0732|
>
> |AVA|Clean|Random|EM|AP|NTGA|AR|REM|SEM|GUE|TUE|MI-UE|
> |-|-|-|-|-|-|-|-|-|-|-|-
> |Test Acc(\%)|89.15|89.00|86.62|88.22|85.13|88.38|86.28|87.30|86.63|88.72|86.18|
> |Acc Gap(\%)|-|0.15|2.53|0.93|4.02|0.77|2.87|1.85|2.52|0.43|2.93|
> |MI|0.7186|0.6600|0.6593|0.6473|0.6329|0.6420|0.6622|0.6397|0.6488|0.6572|0.6238|
> |MI Gap|-|0.0586|0.0593|0.0713|0.0857|0.0766|0.0564|0.0789|0.0698|0.0614|0.0948|
>
> Ref:
>
> [a] Tao et al. Better safe than sorry: Preventing delusive adversaries with adversarial training. NeurIPS 2021.
>
> **Question 2: High computational overhead and unclear scalability…The paper does not report runtime cost, memory overhead, or scalability curves relative to baselines. It would be better to assess such practicality for real-world privacy preservation at large data scale.**
>
> We have provided the computational cost in Appendix G.6, all of our experiments are conducted on a single NVIDIA A800 GPU. Due to incorporation of the similarity matrix of our MI reduction loss, our MI-UE introduce about 1.5 times computational overhead compared with baseline method like EM. An easy and effective approach to mitigate these overheads is to reduce the poisoning epochs. To validate the effectiveness, we test MI-UE with smaller poisoning epochs (50 and 20) for CIFAR-10 and CIFAR-100 in the following table.
>
> |Epochs/Test Acc(\%)    | CIFAR-10 | CIFAR-100|
> |-|-|-|
> |100 (default)  | 9.95 | 1.17 |
> |50  | 10.25 | 1.66 |
> |20  | 15.39 | 3.23 |
>
> Results demonstrate that with half of the generation time (50 epochs), MI-UE still outperforms on CIFAR-10 and achieves the second-best performance on CIFAR-100 compared with existing UEs shown on Table 2 in Section 6. Furthermore, MI-UE still gains comparable results with other UEs even the poisoning epochs are reduced to 20.
>
> Furthermore, for ImageNet-subset, we test our MI-UE with 30 poisoning epochs and 15 poisoning epochs. Compared with Table 2 in Section 6, MI-UE under 30 poisoning epochs demonstrates the state-of-the-art unlearnability across existing UEs. Even for 15 poisoning epochs, MI-UE still achieves the second-best performance, slightly behind EM.
>
> |Epochs/Test Acc(\%)     | ImageNet-subset |
> |-|-|
> |50 (baseline)  | 1.03 |
> |30 | 1.09 |
> |15  | 1.95 |
>
> The effectiveness of MI-UE under economic scenarios further demonstrate MI-UE’s real-world applications. We add these additional discussions on smaller poisoning epochs into our Section 6.4 and Appendix G.6 in our revised manuscript.

---

### Author Response · Authors · 2025-11-24
**Overall Rebuttal**

We thank all reviewers for their valuable feedback and comments. We summarize the revision of our manuscript here.

-	We include additional ablation studies on various training step sizes, different training epochs, and different poisoning budgets on Tables 9 to 11 in Section 6.4.

-	We add defense evaluation on different JPEG compression quality for various unlearnable examples in Section 6.4 and Table 12.

-	We provide the evaluation on linear separability of unlearnable noises and datasets in Appendix G.5 and Table 22.

-	We discuss the rationality of the assumption for Theorem 5.1 in Appendix F.5.

-	We add some discussions on the latest work in Section 2.

-	We make further explanations for some obscure parts in our manuscript to increase the readability.

We hope these modifications can help to address your concerns. Thanks again for your time and professional reviews. We are happy to address further feedback.

Best regards,

Authors of Submission 2203

---

### Author Response · Authors · 2025-12-04
**Summary of Contributions and Rebuttal Responses**

Dear Reviewers and Area Chairs,

We sincerely appreciate the valuable time of all the reviewers and area chairs. **During the rebuttal period, we responded all the reviewers' concerns with detailed clarifications and substantial additional experiments, which we believe resolved their concerns and significantly improved the quality of the paper. Unfortunately, due to the accidental leak of reviewer information, the discussion period was prematurely closed, and although we attempted to engage, the reviewers were unable to provide further feedback or participate in the discussion.**

We understand that this has resulted in a significant workload for the newly assigned AC to evaluate the paper within the limited time available. Below is our comprehensive summary to help clarify the core contributions and key modifications of the paper.

**Significance and Motivation**:

- This paper explores how to more efficiently utilize Unlearnable Examples to protect data privacy and prevent unauthorized deep learning models from learning private data. Existing UE methods rely on empirical heuristics and lack theoretical support. Our paper introduces a novel perspective—mutual information reduction—to improve UEs and provide more effective privacy protection.

**Key Contributions and Strengths**:

- **Our paper is well-written, with clear definitions, logical flow, and accessible explanations.** Complex concepts like MI and covariance reduction are explained systematically. (Reviewer i275)
- **Our approach, idea, and theoretical lens are novel and interesting,** filling a conceptual gap in a field often driven by heuristics/going beyond empirical heuristics. (Reviewers uCnz, i275, L5MP)
- **Our paper includes theoretical grounding,** demonstrating a connection between MI reduction and unlearnability. (Reviewers i275)
- **The performance of our method is effective, and broadly transferable across architectures.** (Reviewers uCnz, i275, L5MP)
- **The experimental evaluation is well-designed, extensive and comprehensive across datasets, models, and defense scenarios** (Reviewers uCnz, i275, L5MP)
- **The ablation is good and solid. The analysis of model depth effect is good and useful.** (Reviewers uCnz, L5MP)

**Key Improvements and Clarifications**:

- **Computational overhead**: To address the concerns regarding computational overhead raised by reviewers uCnz and i275, we have provided a comprehensive analysis of the computational cost in Appendix G.6. The MI-UE method incurs approximately 1.5 times the computational cost of baseline methods like EM due to the introduction of the MI reduction loss. A simple and effective way to mitigate these overheads is to reduce the number of poisoning epochs. We conducted validation experiments demonstrating the practical value of reducing poisoning epochs.

- **Additional datasets and defenses**: In response to reviewer i275's concerns regarding datasets, we extended MI-UE to 3D point clouds. Experiments on the ModelNet40 dataset demonstrate the effectiveness of MI-UE in this context. To address reviewer L5MP's concerns regarding ISS defenses, we added new evaluations in Section 6.4 of the revised manuscript.

- **Hyperparameter sensitivity**: To address reviewer nUBW's concerns regarding hyperparameter sensitivity, we performed additional experiments varying the poisoning budgets from 4/255 to 16/255. For the step size in unlearnable poison training, we tested different values such as 0.1/255, 0.2/255 (default), 0.4/255, and 0.8/255. We also tested MI-UE under different numbers of poisoning epochs.

- **Core concept clarifications**: In response to reviewer L5MP's concerns about linear separability, we clarified misunderstandings and added explanations to Figure 1 in the revised manuscript. Additionally, we addressed concerns regarding the Gaussian distribution assumption raised by reviewer nUBW and clarified its rationale. Besides, we have clarified the novelty and significance of this work. In response to reviewer i275's concerns regarding other optimization strategies and theoretical analysis, we have addressed some misunderstandings.

We believe that the revised manuscript and the detailed explanations in our responses have fully addressed the reviewers' concerns, even though, due to system issues, reviewers were unable to provide further feedback. Once again, we sincerely thank all the reviewers and area chairs for their time and valuable guidance.

Sincerely,

The Authors

---

### Meta-Review · Area_Chair_nJGk · 2025-12-02

**Summary:**

This paper hypothesizes that the reason unlearnable examples work is because they minimize the mutual information between the clean and poisoned features.  The authors conduct both empirical and theoretical studies, and they also develop a new method for generating unlearnable examples, outperforming previous approaches.  Reviewers praised the effectiveness of the new method and also praised the new viewpoint as novel and intuitive.  The reviewers pointed out several criticisms: (1) limited defenses, (2) unscalable, (3) the actual proposed attack is extremely simple and could be developed without any of the theory work in the paper, (4) the theory relies on an assumption of Gaussian mixture model data, (5) “always decrease mutual information” and other language is not well justified by correlation of 0.78 between MI gap and Acc Gap, (6) the attack can be defended by JPEG compression.

**Reviewer Concerns:**

(1) was addressed but JPEG ended up being highly effective against the attack.  (2) was addressed but the method is fundamentally more expensive than alternatives.  It’s also not clear if the experiments the authors run would hold up in larger scale settings. (3) is a fair point and was not sufficiently addressed.  (4) I don’t think this is a big problem inherently. (5) Fair point by reviewer, not satisfactorily addressed.  (6) was confirmed to be true by the authors.  In summary, several points raised across multiple reviewers were not well addressed, but the reviewers will still leaning slightly towards acceptance.

**Reviewer Scores:**

The initial reviewer scores were 6, 6, 2, 6. I believe the reviewer who gave a 2 unreasonably compared the work to a paper published after the ICLR submission deadline, and this reviewer’s other points were addressed sufficiently.  Each of the other reviewers did have points which I do not think they would have changed their opinions on.  Therefore, I will assume an average score between 5.5 and 6.

---

### Decision · Program_Chairs · 2026-01-26

Accept (Poster)